# DIFFUSE AND STEER: CORRECTIVE SAMPLING FOR STABLE 3D MOLECULAR GENERATION

## ABSTRACT

Diffusion models have achieved state-of-the-art performance across diverse domains, yet their application to molecular generation remains challenging. Unlike many data types where values can tolerate slight variations, such as pixel intensities in images, molecules are governed by strict geometric and chemical constraints: minor variations in the atomic coordinates of even a single atom can lead to totally invalid or unstable molecules. These constraints give rise to *highly concentrated* data distributions, forming sharp probability peaks. Moreover, these peaks are *densely packed* in configuration space: changing one atom's type, along with small but precise adjustments to its position and that of its neighbors, can result in a distinct molecule, whereas images generally require much larger perturbations to change semantic meaning. This dense-concentrated structure makes diffusion modeling fragile: because valid regions are narrow and tightly clustered, even small deviations at intermediate timesteps can easily cross validity boundaries. Once entering the invalid regions, the generative process provides unreliable guidance, causing errors that accumulate over timesteps and drift generative trajectories off-distribution, ultimately leading to irreparable structural violations. To address this challenge, we formalize the notion of dense-concentrated structure in molecular distributions and analyze how discrepancies at intermediate steps propagate under reverse inference. Building on this insight, we propose **DIST**, a plug-in corrective method that **DI**ffuses and **ST**eers the intermediate distribution, thereby realigning inference trajectories toward a valid molecular distribution. Our method is model-agnostic and can be integrated into a wide range of existing diffusion models, achieving significant improvements in performance while reducing the computational cost to nearly half the standard number of timesteps.

## 1 INTRODUCTION

Generative models are probabilistic frameworks that aim to approximate an underlying data distribution and generate new samples from the learned distribution. By providing a principled approach to learning and sampling from complex, high-dimensional distributions, generative modeling has emerged as a promising paradigm with broad implications for design automation, simulation, and scientific discovery. Recently, diffusion models (DMs) (Ho et al., 2020; Song et al., 2021b) have become a prominent generative paradigm due to their outstanding performance in natural image synthesis and beyond (Song et al., 2020; Rombach et al., 2021; Watson et al., 2023). A DM consists of a forward process and a reverse process. In the forward process, data samples are gradually corrupted by a Markovian noise injection until they become indistinguishable from pure Gaussian noise. The reverse process is parameterized by a neural network, which is trained to approximate the time-reversed dynamics by iteratively denoising the corrupted states. At inference time, the model generates new samples by simulating this learned reverse trajectory, reconstructing structured data from pure noise. Recent work has extended DMs to 3D molecular generation (Hoogeboom et al., 2022; Xu et al., 2023). However, molecular data presents unique challenges that make direct application of diffusion models less effective.

Specifically, 3D molecules are represented by continuous 3D atomic coordinates together with discrete features such as atom types. Unlike images, where pixel intensities are only loosely correlated and can tolerate a wide range of variations, molecules are governed by strict geometric and chemical constraints, such that even small perturbations to atomic coordinates or atom types can result

in completely invalid or unstable structures (Choi et al., 2025). These constraints result in highly concentrated data distributions with narrow probability peaks, where each peak represents a valid and stable molecular configuration. Even slight displacements can shift the molecular configuration off-peaks into regions of negligible probability, corresponding to invalid or unstable states (Reymond et al., 2012; Martin & Cao, 2015; Bohde et al., 2025). Moreover, these peaks are densely packed but clearly separated: changes in one atom's type, along with small (densely packed) but precise adjustments (well separated) to its position and that of its neighbors, can result in a distinct molecule. **Overall, the molecular distribution exhibits an evident dense and concentrated structure, where each probability peak corresponds to a chemically valid molecule, and the regions between the peaks are of near-zero density.** We provide an illustrative analogy to compare the distribution and diffusion process of images with those of molecules in Fig. 1, to highlight the consequences of such a dense and concentrated structure to the diffusion process. Notably, such denseness breaks the clear supervision signal required for denoising, introduces learning difficulties, and leads to errors that accumulate over time; and because of the concentration of the molecular distribution, such errors cannot be tolerated, ultimately resulting in invalid and unstable generations.

Under the same forward noising process, the peaks of molecular distributions quickly merge creating overlap regions where samples become indistinguishable. In contrast, image distributions exhibit broader peaks that overlap smoothly and only at later stages. However, for the reverse process of molecular diffusion, a critical problem arises: **overlap regions create intersections or crossings of generative trajectories which make the score field inherently ambiguous**, where multiple plausible directions coexist, but the model can only represent a single *averaged* vector. As a result, the learned score is systematically inaccurate in these regions (Liu et al., 2022; Lee et al., 2023; Ni et al., 2025). Because the peaks are thin, discretization error (Zhang et al., 2023), model limitations, and imperfect score estimation in overlap regions can push the reverse updates too far, placing samples into low-density regions (see Fig. 1). The resulting discrepancy between the true data distribution and the model distribution, caused by artificial inflation of probability mass in invalid regions, then accumulates and propagates (Li & van der Schaar, 2023), ultimately leading to irreversible structural failures. We further analyze this phenomenon in Sec. 3.1.

To address this challenge, we focus on the unique nature of molecular data distributions. Since chemically valid molecules occupy only the densely packed distribution peaks, which are confined to narrow and well-separated regions of the representation space, we describe this property as *dense-concentrated structure (DC-structure)*, formally introduced in Definition 3.1 in Sec . 3.1. This definition provides a quantitative handle on the geometry of molecular distributions and lays the theoretical foundation for our analysis. Building on this, we show in Sec. 3.2 how such analysis motivates a corrective method, **DIST**, which **DI**ffuses the intermediate distribution and **ST**eers trajectories back toward valid high-density regions. DIST improves the stability and overall performance of molecular generation, while also providing efficiency gains as an additional benefit.

In this work, our main contributions are:

- **Observation.** We are the first to highlight that molecular data distributions are highly *concentrated* and *dense* that makes diffusion-based generative processes fragile.
- **Theory.** We formalize the notion of DC-structure in molecular distributions and analyze its implications for the intermediate distributions during the diffusion process and error propagation in reverse inference.
- **Method.** Building on this analysis, we design a plug-in corrective module, **DIST**, that can be seamlessly integrated into diverse diffusion-based molecular generation methods.
- **Performance.** Extensive experiments on multiple benchmarks and backbones demonstrate that DIST not only improves stability and overall performance, but also reduces computational cost to nearly half the standard number of timesteps.

## 2 PRELIMINARIES

### 2.1 DIFFUSION MODELS

Diffusion models (DMs) (Ho et al., 2020; Song et al., 2021b) are latent-variable generative models that learn to transform Gaussian noise into data samples through a forward–reverse Markov chain. Let $x \sim p(x)$ denote a clean data sample, and let $z_t$ denote its progressively noised version at timestep $t \in \{0, \ldots, T\}$. Here $T$ is the total number of timesteps, $\beta_t \in (0, 1)$ is a variance-schedule

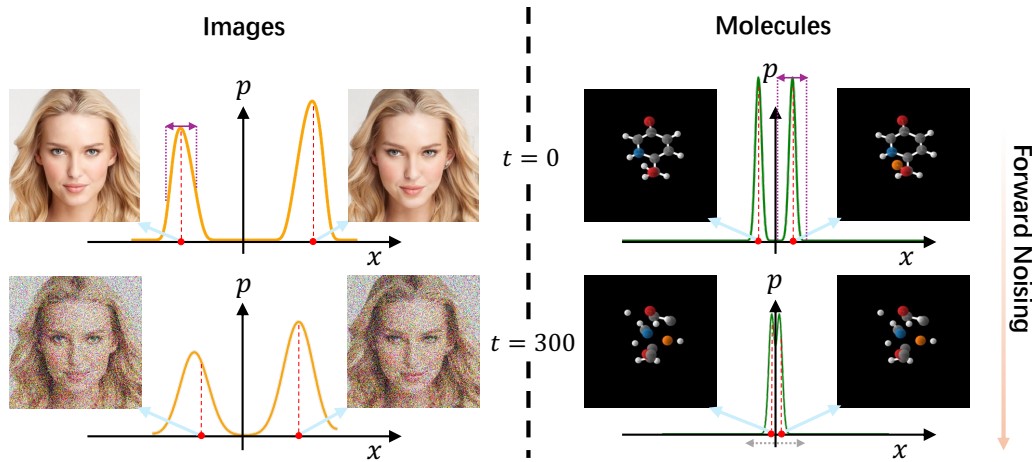

Figure 1: Diffusion process applied to pairs of images (left, obtained from Preechakul et al. (2022)) and molecules (right, extracted from QM9 (Ramakrishnan et al., 2014)) under the same noise schedule. Both pairs start from similar, yet distinct configurations, corresponding to two separate peaks in the distribution. Compared with the image distribution, the molecular distribution is much denser and more concentrated with narrower peaks. Under mild noise corruption at $t = 300$, noisy images remain distinguishable, whereas *noisy molecules quickly become indistinguishable due to the denseness*. At $t = 0$, small errors (indicated in purple) in the image distribution still land in regions of relatively high density, corresponding to visually realistic images, whereas *small errors in the molecular distribution will drift the generated samples into regions of near-zero density between two peaks due to **concentration***.

parameter, and we use the shorthand $\boldsymbol{z}_{1:T} = (\boldsymbol{z}_1, \ldots, \boldsymbol{z}_T)$. The forward process gradually corrupts data by adding Gaussian noise:

$$p(\boldsymbol{z}_{1:T} \mid \boldsymbol{x}) = \prod_{t=1}^{T} p(\boldsymbol{z}_t \mid \boldsymbol{z}_{t-1}), \qquad p(\boldsymbol{z}_t \mid \boldsymbol{z}_{t-1}) = \mathcal{N}\left(\sqrt{1-\beta_t}\,\boldsymbol{z}_{t-1},\, \beta_t I\right). \tag{1}$$

By composition, the marginal conditional distribution admits a closed form:

$$p(\boldsymbol{z}_t \mid \boldsymbol{x}) = \mathcal{N}\left(\sqrt{\bar{\alpha}_t}\,\boldsymbol{x},\, (1-\bar{\alpha}_t)I\right), \quad \bar{\alpha}_t = \prod_{s=1}^{t} \alpha_s = \prod_{s=1}^{t}(1-\beta_s). \tag{2}$$

Here $\alpha_s = 1 - \beta_s$ controls the noising pace (Ho et al., 2020; Nichol & Dhariwal, 2021b). The unconditional marginal at step $t$ is then

$$p(\boldsymbol{z}_t) = \int p(\boldsymbol{x}) \mathcal{N}\left(\boldsymbol{z}_t \,\big|\, \sqrt{\bar{\alpha}_t}\,\boldsymbol{x},\, (1-\bar{\alpha}_t)I\right)\, d\boldsymbol{x}, \tag{3}$$

which interpolates between the data distribution $p(\boldsymbol{x})$ and the Gaussian prior $p(\boldsymbol{z}_T) \approx \mathcal{N}(0, I)$ (Albergo et al., 2023). The reverse process reconstructs data from noise, factorized as $q_\theta(\boldsymbol{z}_{0:T}) = q(\boldsymbol{z}_T) \prod_{t=1}^{T} q_\theta(\boldsymbol{z}_{t-1} \mid \boldsymbol{z}_t)$, with transitions $q_\theta(\boldsymbol{z}_{t-1} \mid \boldsymbol{z}_t) = \mathcal{N}\left(\boldsymbol{\mu}_\theta(\boldsymbol{z}_t, t), \rho_t^2 I\right)$, where $\boldsymbol{\mu}_\theta$ is predicted by a neural network and $\rho_t$ is typically fixed. DMs are trained with the noise-prediction objective (Song et al., 2021b):

$$\mathcal{L}_{\text{DM}} = \mathbb{E}_{\boldsymbol{x},\boldsymbol{\varepsilon},t}\left[\|\boldsymbol{\varepsilon} - \boldsymbol{\varepsilon}_\theta(\boldsymbol{z}_t, t)\|^2\right], \qquad \boldsymbol{z}_t = \sqrt{\bar{\alpha}_t}\,\boldsymbol{x} + \sqrt{1-\bar{\alpha}_t}\,\boldsymbol{\varepsilon}, \quad \boldsymbol{\varepsilon} \sim \mathcal{N}(0, I). \tag{4}$$

The network $\boldsymbol{\varepsilon}_\theta$ can be interpreted as learning the score field $\nabla_{\boldsymbol{z}_t} \log p(\boldsymbol{z}_t)$ (Song et al., 2021a;b). New samples are generated by starting from pure Gaussian noise $\boldsymbol{z}_T \sim \mathcal{N}(0, I)$ and iteratively applying the reverse update:

$$\boldsymbol{z}_{t-1} = \frac{1}{\sqrt{1-\beta_t}}\left(\boldsymbol{z}_t - \frac{\beta_t}{\sqrt{1-\bar{\alpha}_t}}\,\boldsymbol{\varepsilon}_\theta(\boldsymbol{z}_t, t)\right) + \rho_t \boldsymbol{\varepsilon}, \quad \boldsymbol{\varepsilon} \sim \mathcal{N}(0, I). \tag{5}$$

## 2.2 DMs for Molecular Generation

A 3D molecule with $N$ atoms contains both continuous atomic coordinates and discrete atomic features (Hong et al., 2024). The atomic coordinates are represented as $\boldsymbol{x} = (\boldsymbol{x}_1, \ldots, \boldsymbol{x}_N) \in$

$\mathbb{R}^{N \times 3}$, where each $\boldsymbol{x}_i$ denotes the coordinates of an atom in $\mathbb{R}^3$. The atomic features, such as charges and atom types, are represented as $\boldsymbol{h} = (\boldsymbol{h}_1, \ldots, \boldsymbol{h}_N) \in \mathbb{R}^{N \times d}$. While the atomic features are scalar quantities invariant to translations and rotations ($\mathbb{SE}(3)$-transformations), the coordinates transform equivariantly under these transformations (Thomas et al., 2018; Hoogeboom et al., 2022; Dumitrescu et al., 2024). However, arbitrary $\mathbb{SE}(3)$-transformations of the coordinates can cause issues for standard denoising networks, since a rotated or translated molecule may be perceived as an entirely different sample. To overcome such issues, existing works often design $\mathbb{SE}(3)$-equivariant frameworks to ensure symmetry-awareness. Specifically, translations can be handled by subtracting the centroid of atomic coordinates $\boldsymbol{x}$ to remove translational degrees of freedom (Garcia Satorras et al., 2021; Xu et al., 2022). However, rotations are much complicated and often handled by using carefully designed equivariant neural networks (Hoogeboom et al., 2022; Xu et al., 2023) or by canonicalization (Ding & Hofmann, 2025; Kaba et al., 2023; Rempe et al., 2020).

In addition, the hybrid discrete–continuous nature of molecular data (Dunn & Koes, 2024) introduces unique challenges for generative modeling. Several recent works attempt to address the challenge by learning smoother latent representations (Xu et al., 2023; Ding & Hofmann, 2025; Chen et al., 2025; Luo et al., 2025). These approaches typically employ a VAE-based (Kingma & Welling, 2013) encoder-decoder framework, carrying out the diffusion process in a latent space rather than directly on molecular coordinates and features. While this alleviates some modeling challenges, latent-space methods introduce new sources of approximation error, and discrepancies remain between generated molecules and chemically valid structures. Importantly, the error introduced by the learned score model (see equation 4) is ubiquitous and largely independent of architectural choices (Song et al., 2023; 2024; Joshi et al., 2025); we observe such failures across GNN- and Transformer-based models, as well as in both equivariant and non-equivariant molecular generation methods. Moreover, the discrepancy between the true data marginal distribution and the model distribution grows as errors accumulate across timesteps.

This observation indicates that **performance cannot be guaranteed solely by architectural choices** intended to simplify score-matching (Song et al., 2021b). Instead, it highlights the necessity of correcting inference trajectories at intermediate timesteps in order to reduce distributional discrepancies and thereby improve the stability and validity of generated molecules. Moreover, a detailed discussion on the comparison of our work with corrective method is provided in Appendix B.

## 3 METHOD

In this section, we delve into three key questions: (1) How can the unique structure of molecular distributions, constrained by chemical rules, be formally characterized? (2) What issues arise due to this structure for 3D molecular diffusion models? (3) Can these issues be mitigated through correction? We answer the first two questions by formally investigating the DC-structure of molecular distributions in Sec. 3.1. Building on this insight, we propose DIST together with its theoretical analysis in Sec. 3.2, which addresses the last question.

### 3.1 DENSE-CONCENTRATED STRUCTURE ISSUE

As illustrated in Fig. 1, molecular data distribution over the representation space exhibits an evident DC-structure, where each peak corresponds to a chemically valid molecule, and regions between the peaks are of near-zero density. This contrasts with images, where the pixel values can tolerate a wide range of variations, resulting in wider peaks and smoother transitions. **To rigorously capture this phenomenon and further analyze its implications, we next formalize the DC-structure in probabilistic terms.** Consistent with Sec. 1 and prior work, we denote the true and model marginals by $p(\boldsymbol{z}_t)$ and $q_\theta(\boldsymbol{z}_t)$, respectively. Unless otherwise stated, all analysis in this work is carried out under the molecular setting rather than the universal diffusion machinery. For notational simplicity, we write the true marginal as $p_t$ and also omit the learnable parameter $\theta$ and write the model marginal as $q_t$.

**Definition 3.1** (Dense-concentrated Structure). *There exist $K_0$ centers $\{m_k\}$, a scale $\sigma_* > 0$, a separation $\Delta > 0$, and weights $\{w_k\}$ such that, for the operative noise level $t$,*

$$p_t \simeq \sum_{k=1}^{K_0} w_k \mathcal{N}(m_k, \Sigma_{k,t}), \qquad \Sigma_{k,t} \preceq \sigma_*^2 I, \qquad \|m_k - m_\ell\| \geq \Delta \ (k \neq \ell),$$

*and for each $k$ there exists some $\ell \neq k$ with $\|m_k - m_\ell\| \leq O(\Delta)$, and*

$$p_t\left(\bigcup_{k=1}^{K_0} B(m_k, c\sigma_*)\right) \geq 1 - \delta_t$$

*for some $c > 0$ and small $\delta_t \in [0, 1)$, where $B(m, r) = \{x \in \mathbb{R}^d : \|x - m\| \leq r\}$ denotes the Euclidean ball of radius $r$ centered at $m$.*

Under this definition, $p_t$ is a mixture of narrow peaks $\{B(m_k, c\sigma_*)\}_{\forall k}$ separated by low-density gaps. For molecular data, the parameter $\sigma_*$ is small, reflecting that each valid configuration is concentrated in a narrow neighborhood of configuration space. At moderate timesteps $t$, forward noising smooths these peaks (see equation 2 and equation 3) and creates overlap regions between them, thus a sample $z_t$ may lie in the overlap, close to the midpoint between two peaks (see Fig. 1). In this circumstance, the score field points outward, pushing $z_t$ toward the nearest peak with magnitude $\|\nabla \log p(z_t)\| \sim \frac{\Delta}{\sigma_*^2}$ (Song et al., 2021b), and the reverse update step based on equation 5 is

$$\|z_{t-1} - z_t\|_{\det} \approx \beta_t \cdot \frac{\Delta}{\sigma_*^2}. \tag{6}$$

Because $\sigma_*$ is small for molecules under Definition 3.1, this step can easily overshoot the distribution radius $c\sigma_*$ and land in a low-density area:

$$\beta_t \frac{\Delta}{\sigma_*^2} > c\sigma_* \implies z_{t-1} \notin \bigcup_k B(m_k, c\sigma_*). \tag{7}$$

The derivation and toy examples are provided in Appendix C. In other words, when $z_t$ originates from an overlap region created by forward noising, the reverse step is prone to push it across a thin peak and into a low-density region. Subsequent denoising cannot recover from this drift. For images, by contrast, peaks are broad ($\sigma_*$ is large) and can overlap smoothly, so the condition in equation 7 is rarely triggered.

Consequently, the overshoot mechanism in equation 7, which arises directly from the concentration property in Definition 3.1, explains the fragility of reverse inference. The score field $\nabla \log p_t$ indeed points *toward* high-density peaks; however, because molecular peaks are narrow, the reverse update can step *past* the peak and cross the high-density into the opposite regions. **Once outside the distribution, subsequent updates are driven by the model score $\nabla \log q_t$ in a low-density region where estimation and discretization errors are large (Zhang et al., 2023; Li & van der Schaar, 2023), leading to oscillation or further drift rather than reliable re-entry into the correct peak.** Moreover, Cao et al. (2023) also analyzed this re-entry problem and demonstrated the benefits of stochastic samplers, which further underscores the importance of trajectory correction in SDE simulation. This phenomenon is more obvious in molecular generation due to the DC-structure, and we provide a detailed comparison and explanation specific to molecules in Appendix D.

In practice, discrepancies between the true marginal $p_t$ and the model $q_t$ accumulate across timesteps, and low-density region excursions become effectively unrecoverable. As shown in Table 1, inference quality degrades monotonically with $t$ increasing, reflecting the growing deviation between $p_t$ and $q_t$. This motivates the need for a corrective mechanism at intermediate timesteps to prevent off-distribution drift. An overview of our proposed method, **DI**ffuse and **ST**eer (**DIST**), is illustrated in Fig. 2, and we formalize how DIST selectively realigns $q_t$ with $p_t$ in Sec. 3.2.

Table 1: Effect of starting timestep $t$ on sample quality. $t = 0$ uses clean data; $t = 1000$ starts from pure Gaussian noise (standard diffusion). Intermediate $t$ forms $z_t \sim p(z_t \mid x)$, and then we run $t$ reverse steps for generated results. The experiment setting follows EDM on QM9. Higher numbers are better. Please refer to Sec. 4.1 for further details.

| $t$ | Atom Sta (%) | Mol sta (%) | Valid (%) |
|---|---|---|---|
| 0 | 99.0 | 95.2 | 97.7 |
| 100 | 99.0 | 92.7 | 96.4 |
| 300 | 98.9 | 89.1 | 95.5 |
| 500 | 98.7 | 86.2 | 94.3 |
| 1000 | 98.7 | 82.0 | 91.9 |

### 3.2 Diffuse and Steer

As discussed in Sec. 3.1 above, the unique characteristics of the molecular data distribution lead to severe inference and learning difficulties, such that the learned denoiser can be very inaccurate.

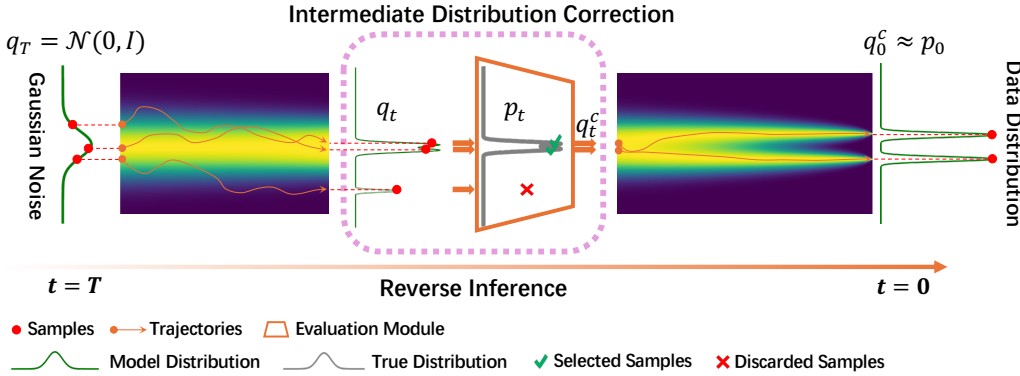

Figure 2: Illustration of **DIST**. In standard reverse inference, trajectories diffuse backward from Gaussian noise $q_T$ toward the data distribution $p_0$, but the model distribution $q_t$ may drift away from the true distribution $p_t$ due to the DC-structure of molecular data (see Sec. 3.1). At an intermediate timestep $t$, **DIST** steers $q_t$ toward $p_t$ via a correction module that evaluates discrepancies and discards invalid samples. The resulting corrected distribution $q_t^c$ better approximates $p_t$, realigning trajectories and improving both stability and validity in the final generation.

As a result, the intermediate model distribution $q_t$ often deviates from the true marginal distribution $p_t$. Moreover, during training, the diffusion model is trained on the true marginal distribution $p_t$ from the dataset. In other words, the reverse process is implicitly learned under the assumption that the intermediate states follow the true marginals. Intuitively, when $q_t$ drifts away from $p_t$, this will create a mismatch between the final distribution obtained by applying the reverse process to $q_t$ and that obtained from $p_t$. Mathematically, we can show that this is true in Corollary 3.1 below.

**Corollary 3.1** (TV–contraction Step). *Let $K_{t\to0}$ be the ideal reverse Markov kernel, which can be intuitively understood as the perfect diffusion model with the true score functions; in other words, when the ideal reverse Markov kernel is applied to the true marginal distribution, we obtain the true data distribution $p_0 = K_{t\to0}p_t$. Then, for any probability measure $q_t$, there exists a TV–contraction coefficient $\kappa \in [0,1]$ such that*

$$\left\|q_0 - p_0\right\|_{\mathrm{TV}} = \left\|K_{t\to0}q_t - K_{t\to0}p_t\right\|_{\mathrm{TV}} \leq \kappa\left\|q_t - p_t\right\|_{\mathrm{TV}}, \tag{8}$$

*where if $q_t$ is the intermediate model distribution, $q_0$ can be understood as the final model distribution obtained by applying the perfect diffusion model on $q_t$.*

The proof and explanation are deferred to the Appendix E.1. **Specifically, Corollary 3.1 reveals that if the intermediate model distribution $q_t$ is closer to the true marginal distribution $p_t$, the final model distribution $q_0$ is closer to the true data distribution $p_0$ that we aim to obtain.** Therefore, to achieve high-quality generation despite the difficulties posed by the molecular data distribution, **our goal is to obtain an improved intermediate distribution $q_t^c$ that remains closer to the true marginal $p_t$ rather than blindly using the model distribution.** To achieve this goal, we propose **DIST** (DIffuse and STeer), a corrective sampling approach for 3D molecular diffusion. Specifically, we perform the reverse process normally as in the standard diffusion pipelines; however, we incorporate an additional correction step to steer the intermediate distribution $q_t$ toward a "corrected" version $q_t^c$ closer to the true marginal $p_t$. An overview of DIST is provided in Fig. 2.

We now present the details of DIST concretely. Building on Definition 3.1, which states that the distribution $p_t$ concentrates around a finite number of peaks separated by low-density regions, we next introduce a finer partition of the support into small neighborhoods. Specifically, we divide the space into radius-$r$ batches $\{B_j\}_{j=1}^J$, which can be regarded as local regions within or around the peaks, each carrying probability mass

$$\pi_j := p_t(B_j), \qquad \hat{\pi}_j := q_t(B_j),$$

together with the conditional distributions $p_{t|j}$ and $q_{t|j}$ restricted to each batch $B_j$.

Each batch $j$ is further associated with a model-side pilot score $s_j \in \mathbb{R}$ (e.g., round-trip residual, self-consistency, ensemble variance, or chemistry-based penalty), which reflects whether the region

is consistent with the true marginal distribution or potentially invalid. Given a threshold $\tau$, we select batches whose scores fall below $\tau$:

$$J^\star(\tau) := \{\, j : s_j \leq \tau \,\}.$$

We then measure how much probability mass remains after this selection by defining

$$\alpha(\tau) := \sum_{j \in J^\star(\tau)} \pi_j, \qquad \beta(\tau) := \sum_{j \in J^\star(\tau)} \hat{\pi}_j.$$

Here, $\alpha(\tau)$ represents the *true coverage*, i.e., the portion of the ground-truth distribution preserved by the selection, while $\beta(\tau)$ denotes the *model coverage*, i.e., the portion of the model distribution retained. Smaller thresholds $\tau$ restrict the selection to batches that are more likely to correspond to valid regions, reducing coverage; larger thresholds broaden the selection and capture more mass, but at the cost of admitting regions inconsistent with the true distribution. The selected model distribution at threshold $\tau$ is then given by

$$q_t^{\mathrm{c}}(\tau) := \sum_{j \in J^\star(\tau)} \tilde{\pi}_j \, q_{t|j}, \qquad \tilde{\pi}_j = \frac{\hat{\pi}_j}{\sum_{k \in J^\star(\tau)} \hat{\pi}_k}. \tag{9}$$

Intuitively, $q_t$ consists of both samples consistent with $p_t$, lying within valid regions, and samples that fall outside. The corrected distribution $q_t^{\mathrm{c}}$ acts as a filtered version of $q_t$, removing invalid batches in order to improve approximation of the true distribution. The following proposition establishes a quantitative error bound that illustrates the effectiveness of DIST.

**Proposition 3.1** (Selective Reverse Error Bound). *Under the DC-structure in Definition 3.1 and the batch construction described above, for any threshold $\tau$ the deviation between the selectively corrected reverse distribution $K_{t \to 0} q_t^{\mathrm{c}}(\tau)$ and the true distribution $p = K_{t \to 0} p_t$ admits an upper bound of the form*

$$\left\| K_{t \to 0} q_t^{\mathrm{c}}(\tau) - p \right\|_{\mathrm{TV}} \;\leq\; f\big(\alpha(\tau), \beta(\tau), (\pi_j, \hat{\pi}_j)_{j \in J^\star(\tau)}, \sup_{j \in J^\star(\tau)} \mathrm{TV}(q_{t|j}, p_{t|j})\big),$$

*where $f(\cdot)$ is an explicit function of the true coverage $\alpha(\tau)$, the model coverage $\beta(\tau)$, the selected batch weights, and the conditional discrepancies. The exact form of $f(\cdot)$ is provided in Appendix E.2.*

The proof and explanation are provided in Appendix E.2. This error bound provides a theoretical guarantee for DIST; that is, **selective correction ensures that $q_t^{\mathrm{c}}$ is steered toward convergence with the true distribution $p$ at intermediate timestep $t$, stabilizing the sampling trajectory.**

**Corrective Sampling**    We now describe how the corrected distribution $q_t^{\mathrm{c}}$ is achieved in the reverse inference procedure (see Fig. 2). At a given intermediate timestep $t$, DIST constructs a candidate pool by reverse-simulating a small set of samples from Gaussian noise at $T$. Each candidate is duplicated and perturbed with a sufficiently small amount of noise to form batches $\{B_j\}_{j=1}^J$, which collectively follow the model distribution $q_t$ and remain within the prescribed radius-$r$ constraint (see Definition 3.1). To evaluate whether these batches $\{B_j\}_{j=1}^J$ are consistent with the true distribution $p_t$, DIST runs a full reverse inference on a pilot subset $\{B_j^{\mathrm{sub}} \mid B_j^{\mathrm{sub}} \in B_j\}_{j=1}^J$ drawn from each batch. This pilot inference provides an empirical assessment of how well the current model trajectory aligns with $p_t$, and serves as a diagnostic of potential drift away from the true distribution. Based on the pilot outcomes $s_j \in \mathbb{R}$, DIST applies a filter $\tilde{\pi}_j$ to each batch using a universal threshold $\tau$, obtaining a corrected distribution $q_t^{\mathrm{c}}(\tau)$ (see equation 9) that better approximates $p_t$. In effect, $q_t^{\mathrm{c}}$ concentrates the reverse trajectories around valid molecular peaks. Beyond improved approximation quality, DIST also provides an efficiency advantage by reducing unnecessary inference on invalid regions, as demonstrated in Sec. 4.3.

## 4 EXPERIMENTS

### 4.1 SETUPS

**Datasets**    Following prior work (Hoogeboom et al., 2022; Xu et al., 2023; Song et al., 2024), we evaluate DIST on two widely used datasets in molecular generation: QM9 (Ramakrishnan et al., 2014) and GEOM-Drugs (Axelrod & Gómez-Bombarelli, 2022). QM9 contains 130K small

Table 2: Results for atom stability, molecule stability, validity, and validity×uniqueness. Higher values indicate better performance. Check Sec. 4.1 for experimental setup details. All models combined with DIST surpass their original counterparts, and the improved results are shown in **bold**. Global best results are underlined.

| # Metrics | QM9 | | | | GEOM-Drugs | |
|---|---|---|---|---|---|---|
| | Atom Sta (%) | Mol Sta (%) | Valid (%) | Valid×Unique (%) | Atom Sta (%) | Valid (%) |
| Data | 99.0 | 95.2 | 97.7 | 97.7 | 86.5 | 99.9 |
| ENF | 85.0 | 4.9 | 40.2 | 39.4 | - | - |
| G-SchNet | 95.7 | 68.1 | 85.5 | 80.3 | - | - |
| EDM | 98.7 | 82.0 | 91.9 | 90.7 | 81.3 | 92.6 |
| **EDM+DIST** | **99.2±0.0** | **89.9±0.3** | **96.9±0.2** | **94.1±0.3** | **82.2** | **96.0** |
| GeoLDM | 98.9 | 89.4 | 93.8 | 92.7 | 84.4 | 99.3 |
| **GeoLDM+DIST** | **99.4±0.0** | **93.4±0.3** | **96.3±0.2** | **93.1±0.2** | **85.4** | **99.7** |
| RADM | 98.5 | 87.3 | 94.1 | 91.7 | 85.0 | 99.3 |
| **RADM+DIST** | **99.1±0.0** | **91.4±0.3** | **96.2±0.1** | **92.3±0.4** | **86.0** | **99.8** |

molecules, restricted to at most 9 heavy atoms (29 atoms including hydrogen atoms). We follow the standard partition from Hoogeboom et al. (2022), with 100K molecules for training, 18K for validation, and 13K for testing. GEOM-Drugs is substantially larger, comprising 420K molecules with an average of 44.4 atoms and up to 181 atoms. Following Hoogeboom et al. (2022), we retain the 30 lowest-energy conformations for each molecule.

**Metrics** Consistent with prior work, we evaluate generated molecules using the following metrics: atom stability, molecule stability, validity, and validity×uniqueness (Simonovsky & Komodakis, 2018; Garcia Satorras et al., 2021). *Atom Stability*: the percentage of atoms whose number of bonds matches their valence (e.g., H:1, C:4, O:2). *Molecule Stability*: the percentage of molecules in which all atoms are stable. *Validity*: the percentage of molecules satisfying valence rules for all atoms. *Uniqueness*: the percentage of molecules that are distinct from one another. Note for GEOM-Drugs, following prior work, we omit the stability and uniqueness metrics, since they are consistently close 0% and 100%, respectively, for all evaluated methods including the baseline methods.

**Baselines** We employ several representative state-of-the-art diffusion models for 3D molecular generation, including EDM (Hoogeboom et al., 2022), GeoLDM (Xu et al., 2023), and RADM$_{DiT-B}$ (Ding & Hofmann, 2025), as backbone models for our proposed DIST and compare with the original without DIST. These backbone diffusion models cover a range of model types, including GNN-based or Transformer-based, equivariant and non-equivariant, and those operating in regular space and latent space. In addition, we include comparisons with well-known non-diffusion-based models, such as ENF (Garcia Satorras et al., 2021) and G-SchNet (Gebauer et al., 2019). The results of backbone models and baseline methods are directly obtained from their original work.

**Implementation Details** To demonstrate the plug-in capability of our DIST and ensure fair comparison, for all backbone models, we strictly use the officially released model weights without altering any hyperparameters or settings for noise schedule, encoder-decoder configurations, and dataset partition. For detailed settings of DIST, please refer to Appendix F.

### 4.2 MAIN RESULTS AND ANALYSIS

To evaluate the performance of each model on QM9 and GEOM-Drug, following prior work, we generate 10,000 3D molecules using each model. The main results are summarized in Table 2. For QM9 dataset, we report averages over three runs together with standard deviations. Across both datasets and all metrics, every backbone model combined with DIST consistently outperforms its original counterpart. The improvements are significant and universal: **all bold numbers in Table 2 indicate that DIST significantly improves the quality of generated molecules, with particularly large margins observed on the most critical stability metrics.** In addition, methods based on our DIST set the new state-of-the-art for molecular generation on both QM9 and GEOM-Drug datasets.

Notably, the margins of improvement observed before and after applying our method highlight the generality of DC-structure issue. Across GNN-based equivariant EDM (Hoogeboom et al., 2022), GeoLDM (Xu et al., 2023) and Transformer-based non-equivariant RADM (Ding & Hofmann, 2025), where GeoLDM and RADM perform in latent space, the issue remains consistently

evident. This observation cautions against relying solely on architectural choices. Our experimental results confirm that, as a plug-in component, DIST effectively steers inference trajectories and thus mitigates distributional discrepancies in the sampling process, providing a valuable complement to architectural innovations to improve 3D molecular generation quality.

## 4.3 EFFICIENCY ANALYSIS

Since the batches $\{B_j\}_{j=1}^J$ are created by duplication and perturbation, DIST requires only $\frac{T-t}{|B|}$ expected timesteps per inference from $T$ (1000 is adopted in backbone models) to $t$, where $|B|$ is the batch size. For example, setting $t = 300$ with $|B| = 100$, each accepted batch after threshold filtering requires only 307 ($\frac{1000-300}{100} + 300$) steps instead of the 1000 steps as used in standard counterparts. A detailed comparison of efficiency is provided in Table 3, which shows DIST can substantially reduce the overall timestep by nearly **half** compared to baselines, while significantly improving the generation quality as shown in Table 2. We also provide a detailed quantification of the expected computational cost of our DIST in Appendix G.1.

## 4.4 ABLATION STUDY

The number of pilot samples drawn from each batch plays a critical role. A larger set of pilot samples provides a more accurate representation of the model distribution $q_t$, and leads to a better corrected distribution $q_t^c$ by DIST. However, increasing the number of pilot samples also leads to higher computational costs. In practice, we may choose a pilot set size that is sufficiently representative while remaining computationally affordable. We conduct an ablation study to compare the final sample quality and computational costs under different numbers of pilot samples, with results reported in Table 4. As expected, increasing the number of pilot samples improves the quality of generated molecules monotonically. At the same time, computational costs (measured by the number

Table 3: Average number of timesteps required for a full inference procedure. The values are computed from the total timestep consumption needed to generate 10,000 molecules, corresponding to the experiments in Table 2. All baseline methods use the standard 1000-step schedule, whereas DIST significantly reduces the computational cost.

| Methods | QM9 | GEOM-Drugs |
|---|---|---|
| EDM+DIST | 556.1 | 503.3 |
| GeoLDM+DIST | 416.9 | 636.7 |
| RADM+DIST | 413.7 | 438.8 |
| Baselines | 1000 | 1000 |

of time steps) also increase monotonically. Nevertheless, even under a relatively small budget (30, 50, 100), DIST still demonstrates superior performance, significantly improving the original EDM in both sample quality and computational efficiency. Moreover, we also constructed the ablation study on hyperparameters, including batch score threshold, intermediate timestep, and perturbation intensity, as shown in Appendix H.

Table 4: Ablation study for varying pilot subset sizes using EDM+DIST on QM9 with a fixed batch size of 100. We report the generation performance and the average number of timesteps.

| Size | Atom Sta (%) | Mol Sta (%) | Valid (%) | Valid×Unique (%) | Timesteps |
|---|---|---|---|---|---|
| 30 | 99.2 | 89.5 | 96.7 | 94.3 | 428.3 |
| 50 | 99.2 | 89.9 | 96.9 | 94.1 | 556.1 |
| 100 | 99.3 | 90.5 | 97.3 | 94.9 | 644.7 |

## 5 CONCLUSION AND FUTURE WORK

In this work, we investigated the unique challenge of applying diffusion models to molecular generation. Molecular data are confined to concentrated regions of the representation space, with chemically valid structures corresponding to densely packed sharp peaks separated by regions of near-zero density. This DC-structure makes diffusion modeling fragile, since small errors at intermediate timesteps are amplified, causing generative trajectories to drift off-distribution and accumulate irreparable structural violations. To address this issue, we proposed DIST, which is a selective correction method that filters and rescales intermediate distributions, steering the inference trajectories toward valid molecular peaks. DIST is model-agnostic and can be integrated into a wide range of

diffusion-based molecular generators. We also provided both theoretical analysis and experimental results to demonstrate that our method consistently improves the performance across multiple architectures for molecular generation, while nearly halving the inference cost.

Looking forward, our work opens several promising directions. First, as a general and principled framework, DIST can be extended to other data domains with a similar distribution structure. An intriguing question is whether the DIST framework can be adapted to protein generation, although this constitutes a fundamentally different and substantially more complex task. Second, adaptive selection or other strategies for filtering may further improve correction efficiency. Finally, while our study focuses on diffusion models, the DC-structure issue is not exclusive to them. Exploring analogous corrective strategies in alternative generative paradigms, such as normalizing flows (Rezende & Mohamed, 2015), autoregressive models (Li et al., 2024), or energy-based frameworks (Du & Mordatch, 2019), may broaden the impact of our approach and provide a unifying principle for modeling highly constrained distributions.

ETHICS STATEMENT

This work adheres to general ethical principles of scientific research. Our goal is to contribute to society and scientific progress by improving generative modeling for molecular data. We have carefully considered possible harms: our method is purely methodological and does not involve sensitive personal data, human subjects, or confidential information. All experiments rely on publicly available molecular datasets, and no privacy concerns arise. We believe our work will benefit the community as a complementary tool for advancing generative modeling, without introducing foreseeable risks of discrimination or misuse beyond the general risks associated with generative models.

REPRODUCIBILITY STATEMENT

All theoretical results are stated with explicit assumptions, and complete proofs are provided in Appendix E.1 and Appendix E.2. The datasets used in our experiments (QM9 and GEOM-Drugs) are publicly available, and we describe all preprocessing steps in Sec. 4.1. After acceptance, we will publicly release the code and provide detailed guidance to facilitate reproduction of all results.

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

## A    THE USE OF LARGE LANGUAGE MODELS (LLMS)

LLMs are used solely to assist with grammar checking and improving writing fluency. All observations, ideas, methodologies, and contributions in this paper are developed entirely by the authors. Any content generated with the help of LLMs is created under detailed author instructions and thoroughly verified by the authors before inclusion.

## B    COMPARISON WITH RELATED METHODS

In this section, we present a focused comparison between DIST and analogous works in both theoretical formulation and corrective sampling techniques.

One innovation of DIST lies in modeling the molecular distribution within the diffusion process as a DC-structure (Definition 3.1). This idea is conceptually related to the notion of a supervision region introduced in the recent work (Song et al., 2025). However, that work interprets the phenomenon mainly as a factor affecting the generalization ability of diffusion trajectories, without addressing its impact on constrained and complex data types such as molecular structures, nor proposing any corrective mechanism. In contrast, DIST explicitly quantifies and mitigates this issue through a theoretically grounded correction process.

To the best of our knowledge, there is currently **no existing corrective method that *directly* steers intermediate distributions in diffusion-based molecular generation**. Two recent studies in the text-to-image domain, Dynamic CFG (Papalampidi et al., 2025) and ELECT (Kim et al., 2025), adopt selective sampling strategies that are loosely related to our idea. Dynamic CFG performs a greedy search to select the scale that maximizes evaluation scores at each step, while ELECT selects the best candidate from a pool at an intermediate timestep and denoises it as the final output. However, both works focus on sample-level quality refinement rather than distributional correction, lack theoretical grounding, and incur additional computational overhead due to repeated candidate discarding.

A separate line of recent work investigates exposure-bias effects in diffusion models (Ning et al., 2023; Wang et al., 2025; Li et al., 2023). These methods primarily analyze the general training–inference mismatch issue that arises during denoising: they study how prediction errors accumulate across reverse steps and how to make individual transitions more stable. Such analyses focus on *local transition dynamics*, such as characterizing variance inflation from prediction (Ning et al., 2023), identifying consistent neighborhoods of training samples (Wang et al., 2025), or aligning adjacent timesteps to reduce mismatch (Li et al., 2023). In contrast, DIST is motivated from a *distribution-level* perspective. Rather than attributing instability to step-wise prediction errors alone, DIST formalizes the dense-concentrated structure of molecular data and identifies it as a fundamental source of fragility in molecular diffusion. The theoretical results of DIST therefore target *global distributional correctness*: by correcting the intermediate model distribution, DIST provides guarantees on the quality of the final distribution at $t = 0$. Methodologically, exposure-bias approaches typically require modified training objectives, adversarial components, or adjusted sampling schedules, whereas DIST introduces a *training-free, plug-in selective correction module* that can be applied directly at inference time and achieves NFE-level efficiency gains without altering the backbone model.

In summary, corrective methods for diffusion generation remain an emerging and promising research direction. DIST contributes to this direction by: (1) introducing the first corrective module tailored for molecular diffusion generation; (2) establishing a solid theoretical foundation for it; and (3) incorporating an explicitly designed efficiency strategy that yields consistent improvements in both generative performance and computational cost over existing baselines.

## C    OVERSHOOT MECHANISM

In this section, we provide the derivation of overshoot mechanism in Appendix C.1 and toy examples for it in Appendix C.2.

### C.1    UPDATE ISSUE

We justify the scaling of score magnitude in the overlap region and reverse update length in equation 6 and equation 7 of the main text. Under the DC-structure (see Definition 3.1),

$$p_t \simeq \sum_{k=1}^{K_0} w_k \mathcal{N}(m_k, \Sigma_{k,t}), \qquad \Sigma_{k,t} \preceq \sigma_*^2 I, \qquad \|m_k - m_\ell\| \geq \Delta \ (k \neq \ell),$$

and define the responsibilities

$$\gamma_k(z_t) = \frac{w_k \mathcal{N}(z_t; m_k, \sigma_*^2 I)}{\sum_j w_j \mathcal{N}(z_t; m_j, \sigma_*^2 I)}.$$

Then

$$\nabla \log p(z_t) = \frac{1}{\sigma_*^2} \Big( \sum_k \gamma_k(z_t) \, m_k - z_t \Big).$$

In a region where $z_t$ is influenced mainly by one or two nearby peaks, the mixture score $\nabla \log p(z_t)$ has the same order of magnitude as the score of the dominant Gaussian components. By Definition 3.1, different peaks are separated by at least $\Delta$, so along directions between two centers $m_k$ and $m_\ell$ we have $\|z_t - m_k\| = \Theta(\Delta)$ in the overlap region visualized in Fig. 1. Hence, up to a constant factor,

$$\|\nabla \log p(z_t)\| \ \sim \ \frac{\Delta}{\sigma_*^2}, \tag{10}$$

which is the scaling used in the main text. And based on the reverse update step used in main text (see equation 5), the deterministic displacement satisfies

$$\|z_{t-1} - z_t\|_{\det} \ \approx \ \beta_t \, \|\nabla \log p(z_t)\|.$$

The detailed derivation is provided in Appendix C.3. Combining with equation 10, we derive $\|z_{t-1} - z_t\|_{\det} \ \approx \ \beta_t \cdot \frac{\Delta}{\sigma_*^2}$. As the radius (see Definition 3.1) is only $c\sigma_*$, when the update step satisfies $\beta_t \frac{\Delta}{\sigma_*^2} > c\sigma_*$ may result in overshooting the distribution $z_{t-1} \notin \bigcup_k B(m_k, c\sigma_*)$.

## C.2 TOY EXAMPLES

To further illustrate the effect of DC-structure (see Definition 3.1) on diffusion sampling, we provide two controlled toy experiments based on Mixture-of-Gaussians (MoG): one in 2D and one in 1D. In each case, we construct *two distributions under identical diffusion settings* (same noise schedule, network architecture, optimizer, and sampling steps): (i) a smooth MoG with well-separated modes, and (ii) a DC-structured MoG with narrow and closely packed peaks. Brighter colors in the heatmaps correspond to higher density regions.

**2D MoG: Effect of DC-structure.** The DC-structured MoG ('Narrow multi-peak MoG' in Fig. 3) contains sharply concentrated peaks placed in close proximity, mimicking the clustered geometric modes observed in molecular data. As shown in Fig. 3, under identical diffusion parameters, sampling from this DC-structured distribution exhibits noticeably poorer behavior: a substantial portion of generated samples drift into low-density regions. This demonstrates that DC-structure imposes additional instability on the reverse diffusion process, even in simple synthetic settings.



Figure 3: 2D Mixture-of-Gaussians examples. Left: smooth MoG. Right: DC-structured MoG.

**1D MoG: Overshoot phenomenon.** The 1D DC-structured MoG ('Sharp 4-peak mixture' in Fig. 4) contains four narrow peaks in close proximity. As illustrated in Fig. 4, at intermediate timesteps (around 25%–50% of the reverse process before generated results), some trajectories

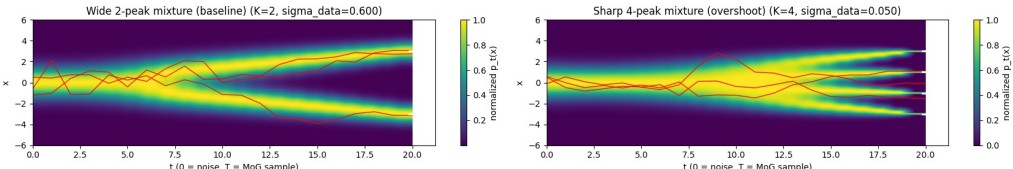

Figure 4: 1D Mixture-of-Gaussians examples. Left: smooth MoG. Right: DC-structured MoG exhibiting overshoot.

*overshoot*: they cross peak regions and fall into low-density regions, ultimately producing invalid samples. This behavior aligns with the overshoot mechanism analyzed in Sec. 3.1.

These synthetic experiments empirically support our theoretical analysis: *DC-structured distributions are significantly more fragile under diffusion sampling*, consistently exhibiting drift, overshoot, and degraded sample quality. This behavior directly parallels challenges found in molecular diffusion models and further motivates the formulation of DIST as a corrective sampling framework for DC-structured data.

### C.3 Deterministic Displacement

In order to analyze the overshoot behavior induced by the DC-structure of $p(\boldsymbol{z}_t)$, it is necessary to isolate the *deterministic displacement* of the reverse update in equation 5. The stochastic noise term is not included in this analysis, because an additive Gaussian perturbation can move a trajectory in an arbitrary direction and therefore obscures the geometric effect we seek to characterize. Our interest is in the model-induced geometric drift of the reverse process, not in the random diffusive fluctuations. For this reason, we study the deterministic quantity $\|\boldsymbol{z}_{t-1} - \boldsymbol{z}_t\|_{\mathrm{det}}$, which captures the intrinsic drift responsible for overshoot, and we provide a detailed derivation below.

Isolate the deterministic displacement of the reverse DDPM update used in equation 5:

$$\boldsymbol{z}_{t-1} = \frac{1}{\sqrt{1-\beta_t}}\Big(\boldsymbol{z}_t - \frac{\beta_t}{\sqrt{1-\bar{\alpha}_t}}\,\boldsymbol{\varepsilon}_\theta(\boldsymbol{z}_t, t)\Big).$$

Subtracting $\boldsymbol{z}_t$ and regrouping terms gives

$$\boldsymbol{z}_{t-1} - \boldsymbol{z}_t = \frac{1}{\sqrt{1-\beta_t}}\,\boldsymbol{z}_t - \frac{\beta_t}{\sqrt{1-\beta_t}\sqrt{1-\bar{\alpha}_t}}\,\boldsymbol{\varepsilon}_\theta(\boldsymbol{z}_t, t) - \boldsymbol{z}_t$$

$$= \underbrace{\Big(\frac{1}{\sqrt{1-\beta_t}} - 1\Big)}_{A_t}\boldsymbol{z}_t \ - \ \underbrace{\frac{\beta_t}{\sqrt{1-\beta_t}\sqrt{1-\bar{\alpha}_t}}}_{B_t}\,\boldsymbol{\varepsilon}_\theta(\boldsymbol{z}_t, t).$$

Thus the coefficients in front of $\boldsymbol{z}_t$ and the score term are, respectively,

$$A_t = \frac{1}{\sqrt{1-\beta_t}} - 1 = \frac{1-\sqrt{1-\beta_t}}{\sqrt{1-\beta_t}}, \qquad B_t = \frac{\beta_t}{\sqrt{1-\beta_t}\sqrt{1-\bar{\alpha}_t}}.$$

EDM (Hoogeboom et al., 2022) and subsequent works (Xu et al., 2023; Ding & Hofmann, 2025) adopt the cosine noise schedule (Nichol & Dhariwal, 2021a), under which $\beta_{300} \approx 1.6 \times 10^{-3}$. This confirms that $\beta_t$ remains very small at moderate timesteps.

**Relative scale of $\boldsymbol{z}_t$ and $\boldsymbol{\varepsilon}_\theta$.** During training, under the forward process (see equation 4), both $\boldsymbol{z}_0$ and $\boldsymbol{\varepsilon}$ have $O(1)$ variance. Thus for typical samples

$$\|\boldsymbol{z}_t\| \ \asymp \ \|\boldsymbol{\varepsilon}\| \ \asymp \ \|\boldsymbol{\varepsilon}_\theta(\boldsymbol{z}_t, t)\|,$$

since the DDPM noise-prediction objective ensures that $\boldsymbol{\varepsilon}_\theta(\boldsymbol{z}_t, t)$ matches the distribution of $\boldsymbol{\varepsilon}$. Hence

$$\|A_t\boldsymbol{z}_t\| \asymp |A_t|, \qquad \|B_t\boldsymbol{\varepsilon}_\theta(\boldsymbol{z}_t, t)\| \asymp |B_t|.$$

**Asymptotic dominance of the score-driven transport term.** A Taylor expansion gives

$$A_t = \frac{\beta_t}{2} + O(\beta_t^2), \qquad \frac{1}{\sqrt{1-\beta_t}} = 1 + O(\beta_t),$$

so

$$B_t = \frac{\beta_t}{\sqrt{1-\bar{\alpha}_t}}\big(1 + O(\beta_t)\big) \asymp \frac{\beta_t}{\sqrt{1-\bar{\alpha}_t}}.$$

Therefore

$$\frac{\|A_t \boldsymbol{z}_t\|}{\|B_t \boldsymbol{\varepsilon}_\theta(\boldsymbol{z}_t, t)\|} \asymp \frac{|A_t|}{|B_t|} = \frac{\frac{\beta_t}{2} + O(\beta_t^2)}{\beta_t/\sqrt{1-\bar{\alpha}_t}} = \frac{1}{2}\sqrt{1-\bar{\alpha}_t} + O(\beta_t).$$

Since $1 - \bar{\alpha}_t \to 0$ as $t \to 0$, we obtain the limit

$$\lim_{t\to 0} \frac{\|A_t \boldsymbol{z}_t\|}{\|B_t \boldsymbol{\varepsilon}_\theta(\boldsymbol{z}_t, t)\|} = 0,$$

i.e. the Gaussian contraction drift is asymptotically negligible compared to the score-driven transport term at a moderate $t$.

**Final approximation.** Using the identity

$$\nabla \log p(\boldsymbol{z}_t) = -\frac{1}{\sqrt{1-\bar{\alpha}_t}}\boldsymbol{\varepsilon}_\theta(\boldsymbol{z}_t, t),$$

the $1/\sqrt{1-\bar{\alpha}_t}$ factor is absorbed into the score parameterization, so that $B_t \boldsymbol{\varepsilon}_\theta$ becomes a coefficient of order $\beta_t$ in front of $\nabla \log p(\boldsymbol{z}_t)$. Thus the deterministic displacement satisfies

$$\|\boldsymbol{z}_{t-1} - \boldsymbol{z}_t\|_{\mathrm{det}} \approx \beta_t \|\nabla \log p(\boldsymbol{z}_t)\|,$$

with the Gaussian contraction drift $A_t \boldsymbol{z}_t$ being strictly lower order in the small-step regime.

## D  SUPPLEMENTARY EXPERIMENTS

The experiment shown in Table 5 is designed to illustrate how stochastic correction in diffusion sampling interacts with the DC-structure of molecular data. In molecular distributions characterized by densely packed and sharply concentrated peaks (Sec. 3.1), even small prediction errors can cause sampling trajectories to drift into low-density regions. SDE–based samplers such as DDPM (Ho et al., 2020) introduce random perturbations that help trajectories re-enter valid high-density regions, effectively compensating for such errors (Cao et al., 2023). In contrast, ODE–based samplers like DDIM (Song et al., 2020) are fully deterministic, lacking this corrective mechanism and thus being more susceptible to error accumulation when the distribution is highly concentrated.

To test this hypothesis, we compare the generation behavior of molecular data (with DC-structure) and image data (with smoother distributions) under both DDIM and DDPM sampling. As shown in Table 5, **DDIM consistently outperforms DDPM in image generation (CIFAR10), whereas the opposite trend appears for molecular generation (QM9), where DDPM yields substantially higher molecule stability across all timestep settings**. These results indicate that the stochasticity in DDPM sampling mitigates overshoot and helps realign trajectories with valid molecular peaks. This finding supports our theoretical analysis in Sec. 3.1, which identified DC-structure as a key factor causing fragility in reverse inference, and it further motivates the design of our proposed DIST method, which explicitly corrects intermediate distributions to achieve similar stabilization effects in a principled manner.

## E  PROOFS

Here we provide the proofs of Corollary 3.1 and Proposition 3.1.

### E.1  COROLLARY 3.1

We first give the preliminary knowledge about the following corollary then derive the proof.

Table 5: Comparison between CIFAR10 image generation (measured by FID) and QM9 molecular generation (measured by molecule stability) under DDIM (Song et al., 2020) and DDPM (Ho et al., 2020) sampling. Results are reported for different numbers of timesteps $S$. Lower is better for FID, higher is better for stability. CIFAR10 numbers are taken from the original DDIM paper (Song et al., 2020), and QM9 experiments follow the setup of EDM (Hoogeboom et al., 2022). The better values are shown in **bold**.

| | CIFAR10 FID ↓ | | | | | QM9 Molecule Stability (%) ↑ | | | | |
|---|---|---|---|---|---|---|---|---|---|---|
| $S$ | 10 | 20 | 50 | 100 | 1000 | 10 | 20 | 50 | 100 | 1000 |
| DDIM | **13.36** | **6.84** | **4.67** | **4.16** | **4.04** | 9.6 | 37.4 | 57.0 | 63.0 | 65.9 |
| DDPM | 41.07 | 18.36 | 8.01 | 5.78 | 4.73 | **12.4** | **50.9** | **74.1** | **77.9** | **82.0** |

**Total Variation (TV) distance** For two probability measures $\mu, \nu$ on a measurable space $(X, \mathcal{F})$, the *total variation distance* is defined by

$$\|\mu - \nu\|_{\text{TV}} := \sup_{A \in \mathcal{F}} |\mu(A) - \nu(A)|.$$

Equivalently, by the Hahn–Jordan decomposition, it admits the dual characterization

$$\|\mu - \nu\|_{\text{TV}} = \tfrac{1}{2} \sup_{\|f\|_\infty \leq 1} \Big| \int_X f \, d\mu - \int_X f \, d\nu \Big|,$$

where the supremum is over all measurable test functions $f$ with $\|f\|_\infty \leq 1$.

**Contraction under Markov kernels** Let $K$ be a Markov kernel, i.e. $K(x, \cdot)$ is a probability measure for each $x \in X$, and $x \mapsto K(x, A)$ is measurable for each $A \in \mathcal{F}$. For a probability measure $\mu$, define the pushforward

$$(K\mu)(A) := \int_X K(x, A) \, \mu(dx), \qquad A \in \mathcal{F}.$$

For a bounded measurable function $f : X \to \mathbb{R}$, define

$$(Kf)(x) := \int_X f(y) \, K(x, dy).$$

Note that $|Kf(x)| \leq \|f\|_\infty$, hence $\|Kf\|_\infty \leq \|f\|_\infty$.

**Corollary 3.1** (TV–contraction Step). *Let $K_{t \to 0}$ be the ideal reverse Markov kernel, which can be intuitively understood as the perfect diffusion model with the true score functions; in other words, when the ideal reverse Markov kernel is applied to the true marginal distribution, we obtain the true data distribution $p_0 = K_{t \to 0} p_t$. Then, for any probability measure $q_t$, there exists a TV–contraction coefficient $\kappa \in [0, 1]$ such that*

$$\big\| q_0 - p_0 \big\|_{\text{TV}} = \big\| K_{t \to 0} q_t - K_{t \to 0} p_t \big\|_{\text{TV}} \leq \kappa \big\| q_t - p_t \big\|_{\text{TV}}, \qquad (8)$$

*where if $q_t$ is the intermediate model distribution, $q_0$ can be understood as the final model distribution obtained by applying the perfect diffusion model on $q_t$.*

*Proof.* For any two probability measures $\mu, \nu$,

$$\|K\mu - K\nu\|_{\text{TV}} \leq \|\mu - \nu\|_{\text{TV}},$$

let $\Delta := \mu - \nu$. Use the dual characterization of TV and Fubini's theorem,

$$\|K\mu - K\nu\|_{\text{TV}} = \tfrac{1}{2} \sup_{\|f\|_\infty \leq 1} \Big| \int f \, d(K\Delta) \Big|$$

$$= \tfrac{1}{2} \sup_{\|f\|_\infty \leq 1} \Big| \int (Kf)(x) \, d\Delta(x) \Big|$$

$$\leq \tfrac{1}{2} \sup_{\|g\|_\infty \leq 1} \Big| \int g(x) \, d\Delta(x) \Big| = \|\mu - \nu\|_{\text{TV}},$$

where we set $g = Kf$, which still satisfies $\|g\|_\infty \leq 1$. Let $K := K_{t\to 0}$ and $p := Kp_t$. Applying the above inequality to $q_t, p_t$ gives

$$\|Kq_t - Kp_t\|_{\mathrm{TV}} \leq \|q_t - p_t\|_{\mathrm{TV}}.$$

Thus, we obtain a contraction $\kappa \in [0,1]$. $\qquad\qquad\square$

To sharpen this to a **stricter** inequality, we introduce the Dobrushin (TV) coefficient:

$$\delta(K) := \sup_{x,x'} \|K(x,\cdot) - K(x',\cdot)\|_{\mathrm{TV}} = 1 - \inf_{x,x'} \int \min\{K(x,dy), K(x',dy)\}.$$

Then for all probability measures $\mu, \nu$,

$$\|K\mu - K\nu\|_{\mathrm{TV}} \leq \delta(K)\,\|\mu - \nu\|_{\mathrm{TV}}.$$

Because in reverse diffusion $K_{t\to 0}(x,dy) = \mathcal{N}(y; \mu_t(x), \Sigma_t)\,dy$ has continuous Gaussian densities with full support, the overlap integral satisfies

$$\int \min\{K(x,dy), K(x',dy)\} > 0,$$

which implies $\delta(K) < 1$. Consequently,

$$\|K\mu - K\nu\|_{\mathrm{TV}} < \|\mu - \nu\|_{\mathrm{TV}}, \qquad \kappa < 1.$$

### E.2 PROPOSITION 3.1

Fix a threshold $\tau$ and follow Sec. 3.2, then we have

$$J^\star := J^\star(\tau), \quad A_\tau := \bigcup_{j \in J^\star} B_j, \quad \alpha := \alpha(\tau) = p_t(A_\tau), \quad \beta := \beta(\tau) = q_t(A_\tau).$$

Recall the selected mixtures and weights

$$q_t^{\mathrm{c}} = \sum_{j \in J^\star} \tilde{\pi}_j\, q_{t|j}, \ \ \tilde{\pi}_j = \frac{\hat{\pi}_j}{\beta}, \qquad p_t^{\mathrm{c}} = \sum_{j \in J^\star} \bar{\pi}_j\, p_{t|j}, \ \ \bar{\pi}_j = \frac{\pi_j}{\alpha}.$$

**Proposition 3.1** (Selective Reverse Error Bound). *Under the DC-structure in Definition 3.1 and the batch construction described above, for any threshold $\tau$ the deviation between the selectively corrected reverse distribution $K_{t\to 0}q_t^{\mathrm{c}}(\tau)$ and the true distribution $p = K_{t\to 0}p_t$ admits an upper bound of the form*

$$\left\|K_{t\to 0}q_t^{\mathrm{c}}(\tau) - p\right\|_{\mathrm{TV}} \leq f\big(\alpha(\tau), \beta(\tau), (\pi_j, \hat{\pi}_j)_{j\in J^\star(\tau)}, \sup_{j\in J^\star(\tau)} \mathrm{TV}(q_{t|j}, p_{t|j})\big),$$

*where $f(\cdot)$ is an explicit function of the true coverage $\alpha(\tau)$, the model coverage $\beta(\tau)$, the selected batch weights, and the conditional discrepancies. The exact form of $f(\cdot)$ is provided in Appendix E.2.*

*Proof.* Conditioning $p_t$ on $A_\tau$ drops exactly $1 - \alpha$ true mass, hence **coverage term** is

$$\|p_t^{\mathrm{c}} - p_t\|_{\mathrm{TV}} = 1 - \alpha. \tag{11}$$

Add and subtract $\sum_{j\in J^\star} \tilde{\pi}_j p_{t|j}$ and use the triangle inequality to obtain **selected-region term**:

$$\|q_t^{\mathrm{c}} - p_t^{\mathrm{c}}\|_{\mathrm{TV}} \leq \underbrace{\left\|\sum_{j\in J^\star} \tilde{\pi}_j\,(q_{t|j} - p_{t|j})\right\|_{\mathrm{TV}}}_{\text{component error}} + \underbrace{\left\|\sum_{j\in J^\star} (\tilde{\pi}_j - \bar{\pi}_j)\,p_{t|j}\right\|_{\mathrm{TV}}}_{\text{weight mismatch}}. \tag{12}$$

For the **component term**, TV is convex:

$$\Big\| \sum_j \tilde{\pi}_j \left(q_{t|j} - p_{t|j}\right) \Big\|_{\mathrm{TV}} \le \sum_j \tilde{\pi}_j \|q_{t|j} - p_{t|j}\|_{\mathrm{TV}}. \tag{13}$$

Now separate the shared and unshared parts of the two distributions. Let $c_j := \min\{\tilde{\pi}_j, \bar{\pi}_j\}$ and write $\tilde{\pi}_j = c_j + r_j$, $\bar{\pi}_j = c_j + s_j$ with $r_j, s_j \ge 0$ and $\sum_j r_j = \sum_j s_j = \frac{1}{2}\sum_j |\tilde{\pi}_j - \bar{\pi}_j|$. Then

$$\sum_j \tilde{\pi}_j \left(q_{t|j} - p_{t|j}\right) = \sum_j c_j \left(q_{t|j} - p_{t|j}\right) + \sum_j r_j q_{t|j} - \sum_j s_j p_{t|j}.$$

Using convexity,

$$\Big\| \sum_j \tilde{\pi}_j \left(q_{t|j} - p_{t|j}\right) \Big\|_{\mathrm{TV}} \le \sum_j c_j \|q_{t|j} - p_{t|j}\|_{\mathrm{TV}} + \tfrac{1}{2}\sum_j |\tilde{\pi}_j - \bar{\pi}_j|. \tag{14}$$

For the weight term in equation 12, because $\sum_j (\tilde{\pi}_j - \bar{\pi}_j) = 0$,

$$\Big\| \sum_{j \in J^\star} (\tilde{\pi}_j - \bar{\pi}_j) p_{t|j} \Big\|_{\mathrm{TV}} \le \tfrac{1}{2} \sum_{j \in J^\star} |\tilde{\pi}_j - \bar{\pi}_j|. \tag{15}$$

Combining equation 14 and equation 15, and noting $\sum_j c_j \le 1$, gives

$$\|q_t^{\mathrm{c}} - p_t^{\mathrm{c}}\|_{\mathrm{TV}} \le \sup_{j \in J^\star} \|q_{t|j} - p_{t|j}\|_{\mathrm{TV}} + \|\tilde{\pi} - \bar{\pi}\|_1. \tag{16}$$

Renormalize the weights,

$$\|\tilde{\pi} - \bar{\pi}\|_1 = \sum_{j \in J^\star} \Big| \frac{\hat{\pi}_j}{\beta} - \frac{\pi_j}{\alpha} \Big| = \frac{1}{\alpha\beta} \sum_{j \in J^\star} |\alpha \hat{\pi}_j - \beta \pi_j|$$

$$= \frac{1}{\alpha\beta} \sum_{j \in J^\star} |\alpha(\hat{\pi}_j - \pi_j) + (\alpha - \beta)\pi_j| \le \frac{1}{\alpha\beta} \left( \alpha \sum_{j \in J^\star} |\hat{\pi}_j - \pi_j| + |\alpha - \beta| \sum_{j \in J^\star} \pi_j \right)$$

$$= \frac{\|\hat{\pi} - \pi\|_1}{\beta} + \frac{|\alpha - \beta|}{\beta} \le \frac{2\|\hat{\pi} - \pi\|_1}{\beta} \le \frac{2\|\hat{\pi} - \pi\|_1}{\min\{\alpha, \beta\}}. \tag{17}$$

By symmetry (swapping $\alpha, \beta$) we get $\|\tilde{\pi} - \bar{\pi}\|_1 \le 2\|\hat{\pi} - \pi\|_1/\alpha$, so equation 17 is the uniform bound.

By the triangle inequality with equation 11 and equation 16, the final discrepancy at timestep $t$

$$\|q_t^{\mathrm{c}} - p_t\|_{\mathrm{TV}} \le (1 - \alpha) + \|\tilde{\pi} - \bar{\pi}\|_1 + \sup_{j \in J^\star} \|q_{t|j} - p_{t|j}\|_{\mathrm{TV}}. \tag{17}$$

Using equation 17 we obtain the safe bound

$$\|q_t^{\mathrm{c}} - p_t\|_{\mathrm{TV}} \le (1 - \alpha) + \frac{2\|\hat{\pi} - \pi\|_1}{\min\{\alpha, \beta\}} + \sup_{j \in J^\star} \|q_{t|j} - p_{t|j}\|_{\mathrm{TV}}. \tag{18}$$

Let $K_{t \to 0}$ be the ideal reverse kernel and $p := K_{t \to 0} p_t$. By Appendix E.1 there exists $\kappa \in [0, 1]$ such that

$$\|K_{t \to 0}\mu - K_{t \to 0}\nu\|_{\mathrm{TV}} \le \kappa \|\mu - \nu\|_{\mathrm{TV}} \quad \forall \mu, \nu. \tag{19}$$

Taking $\mu = q_t^{\mathrm{c}}$ and $\nu = p_t$ and combining with equation 18 gives the form of $f(\cdot)$ by:

$$\|K_{t \to 0} q_t^{\mathrm{c}} - p\|_{\mathrm{TV}} \le \kappa \left[ (1 - \alpha) + \frac{2\|\hat{\pi} - \pi\|_1}{\min\{\alpha, \beta\}} \right]. \tag{20}$$

$\square$

Please refer to Appendix E.3 for further details about the confidence bound and estimation error of $\alpha$ and $\beta$.

### E.3 CONFIDENCE BOUND

Even though the intermediate distributions are intractable, some quantities mentioned in Appendix E.2 can be empirically estimated from available samples of both the forward and reverse diffusion processes, and their estimation errors can be rigorously bounded. In diffusion models we can draw i.i.d. samples $z_t$ from $p_t$ and $z'_t$ from $q_t$, we estimate

$$\widehat{\alpha}(\tau) = \frac{1}{n_p} \sum_{i=1}^{n_p} \mathbf{1}\{z_t^{(i)} \in A_\tau\}, \qquad \widehat{\beta}(\tau) = \frac{1}{n_q} \sum_{i=1}^{n_q} \mathbf{1}\{z_t'^{(i)} \in A_\tau\}.$$

And by Hoeffding's inequality,

$$\mathbb{P}(|\widehat{\alpha} - \alpha| \leq \epsilon_\alpha) \geq 1 - \delta_\alpha, \quad \epsilon_\alpha = \sqrt{\frac{\log(2/\delta_\alpha)}{2n_p}},$$

and similarly for $\widehat{\beta}$. We define lower confidence bounds

$$\alpha_L = \max\{\widehat{\alpha} - \epsilon_\alpha, 0\}, \qquad \beta_L = \max\{\widehat{\beta} - \epsilon_\beta, 0\}.$$

A multinomial $L_1$ concentration inequality (Weissman et al., 2003) provides

$$\Pr(\|\hat{\pi} - \pi\|_1 \geq \epsilon) \leq (2^K - 2)\, e^{-m\, \epsilon^2/2},$$

where $K = |J^\star(\tau)|$ and $m$ is the number of samples used to estimate $\pi$. Hence, with probability at least $1 - \delta_\pi$,

$$\|\hat{\pi} - \pi\|_1 \leq \sqrt{\frac{2\log((2^K - 2)/\delta_\pi)}{m}}.$$

Substituting these empirical bounds into equation equation 20 yields the **finite-sample selective reverse error bound**:

$$\left\|K_{t\to 0}q_t^c(\tau) - p\right\|_{\mathrm{TV}} \leq \kappa\left[(1 - \alpha_L) + \frac{2\,U(n_p, \delta_\pi)}{\min\{\alpha_L, \beta_L\}} + \sup_{j \in J^\star(\tau)} \mathrm{TV}(q_{t|j}, p_{t|j})\right],$$

where

$$U(n_p, \delta_\pi) = \sqrt{\frac{2\log((2^K - 2)/\delta_\pi)}{n_p}}.$$

This bound holds with probability at least $1 - (\delta_\alpha + \delta_\beta + \delta_\pi)$. All terms are **computable from samples**: $\alpha_L, \beta_L$ from empirical coverage and $U(n_p, \delta_\pi)$ from batch counts.

## F SETTINGS OF DIST

DIST is applied at an intermediate timestep $t$ to correct the distribution $q_t$. Concretely, when applied to EDM (Hoogeboom et al., 2022) and GeoLDM (Xu et al., 2023), DIST firstly denoises samples from $t = 1000$ to $t = 300$[1], then duplicates and perturbs each sample 100 times to form batches $\{B_j\}_{j=1}^J$ under a radius-$r$ constraint. From each batch, half of the elements are randomly selected as pilot subsets $\{B_j^{\mathrm{sub}} \mid B_j^{\mathrm{sub}} \in B_j\}_{j=1}^J$, and pilot outcomes $s_j \in \mathbb{R}$ are evaluated based on the stability and validity of the final generated molecules. When applied to RADM (Ding & Hofmann, 2025), the settings are the same except for: the pilot subsets on QM9, which are fixed as 30% of the batch size; and the first-stage denoising on GEOM-Drugs, which terminates at $t = 200$.

## G EFFICIENCY QUANTIFICATION OF DIST

### G.1 THEORETICAL TIMESTEP EFFICIENCY

In this section, we quantify the expected computational cost of the DIST. Let $T$ denote the total number of timesteps in standard inference, and let $t < T$ be the intermediate timestep at which we form the candidate pool. Each candidate is then duplicated into batches of size $|B|$, and only

---

[1] All three baselines use a total of $T = 1000$ timesteps.

a fraction $r_{\mathrm{c}}$ are retained after pilot evaluation, which incurs an additional cost $C_{\mathrm{pilot}}$ per sample. Analytically, the expected cost for a valid sample will be:

$$\mathbb{E}[\text{cost}] = \left(\frac{1}{r_{\mathrm{c}}} - 1\right)\left[\frac{T-t}{|B|} + C_{\mathrm{pilot}}\right] + \left[\frac{T-t}{|B|} + t\right]. \tag{21}$$

The first term accounts for the discarded forward propagation and evaluation of discarded batches, while the second term corresponds to the trajectory for selected batches requiring full reverse inference from $t$ to $0$. We now refine the cost expression by modeling the pilot evaluation as proportional to the selection timestep. Specifically, we set

$$C_{\mathrm{pilot}} = \gamma t, \qquad \gamma = \frac{|B^{\mathrm{sub}}|}{|B|}, \tag{22}$$

where $\gamma$ denotes the proportion of elements in a batch used for pilot evaluation (see Sec. 3.2). Substituting this into the expected cost gives

$$\mathbb{E}[\text{cost}] = \left(1 - \gamma + \frac{\gamma}{r_{\mathrm{c}}}\right)t + \frac{T-t}{|B|\,r_{\mathrm{c}}}. \tag{23}$$

Importantly, the expected cost depends on $r_{\mathrm{c}}$, which is correlated with the final generation quality. However, in practice, the total cost is usually much smaller than $T$. For example, in the EDM+DIST experiment on QM9 (as shown in Table 2), the settings are $T = 1000$, $|B| = 100$, and $\gamma = 0.5$. The empirical estimate of $r_{\mathrm{c}}$ is near $0.39$, leading to an empirical cost of $556.1$ steps on average.

Please refer to Appendix G.2 for detailed implementation of DIST and end-to-end wall-clock usage for efficiency.

## G.2   Wall-Clock Efficiency

In this section, we provide a pseudocode implementation of DIST and measure its end-to-end time efficiency using wall-clock runtime. All experiments are conducted on a single NVIDIA RTX A6000 GPU with CUDA 12.4. To ensure a fair comparison and remain consistent with prior work (Hoogeboom et al., 2022; Xu et al., 2023; Ding & Hofmann, 2025), we follow the standard setting of generating 10,000 molecules with a fixed batch size of 100 (see Sec. 4.1).

To better explain the efficiency mechanism and justify the reproducibility of DIST, we present the pseudocode in Algorithm 1. For consistency with the theoretical foundation in Sec. 3.2 and to illustrate the parallelism of the data flow, the inference is performed in a batch-wise manner. In existing works (Hoogeboom et al., 2022; Song et al., 2024; Xu et al., 2023; Ding & Hofmann, 2025), diffusion models generate molecules batch by batch, whereas DIST introduces a novel efficient paradigm that constructs an intermediate model distribution and explicitly evaluates its deviation from the true distribution. This selective correction is the source of the computational savings.

The iterative Markovian denoising procedure is the main bottleneck for speeding up inference in diffusion models (Frans et al., 2024), so the consumption of GPU-intensive timesteps is a key quantity to consider. In addition to the timestep-based analysis in Table 3 and Appendix G.1, we also report the end-to-end wall-clock runtime for DIST combined with the baseline models. As shown in Table 6, DIST consistently improves the efficiency of the backbone methods, in line with the timestep-based results in Table 3 and the theoretical analysis in Appendix G.1.

We would like to clarify that **DIST is designed as a corrective sampling method for stabilizing molecular generation, with its efficiency improvement being an advantageous side effect**. As such, DIST is not directly comparable to diffusion-model acceleration techniques.

## H   Supplementary Ablation Study

This section provides additional ablation studies for the hyperparameters of DIST, including the batch score threshold $\tau$, the intermediate timestep $t$, and the perturbation intensity used in the radius-$r$ assumption, each introduced in Sec. 3.2. For each ablation experiment, we vary only a

---

**Algorithm 1** Diffuse and Steer (DIST)

---

**Inputs:** neural network $\varepsilon_\theta$, target number $N$, intermediate timestep $t^\star$, threshold $\tau$, pilot size $|B^{\mathrm{sub}}|$, pilot score module Eval
**Output:** set of generated molecules $\mathbf{S}$
Initialize $\mathbf{S} \leftarrow \emptyset$
Sample initial noise batch $\{z_T\} \sim \mathcal{N}(\mathbf{0}, \boldsymbol{I})$
**while** $|\mathbf{S}| < N$ **do**
    **for** $t = T, T-1, \ldots, t^\star + 1$ **do**
        Sample $\{\boldsymbol{\varepsilon}\} = \{(\varepsilon_x, \varepsilon_h)\} \sim \mathcal{N}(\mathbf{0}, \boldsymbol{I})$
        Subtract center of mass from $\varepsilon_x$
        $\{z_{t-1}\} \leftarrow \frac{1}{\sqrt{1-\beta_t}}\Big(\{z_t\} - \frac{\beta_t}{\sqrt{1-\bar{\alpha}_t}}\,\varepsilon_\theta(\{z_t\}, t)\Big) + \rho_t\{\boldsymbol{\varepsilon}\}$
    **end for**
    Duplicate and perturb $\{z_{t^\star}\}$ to obtain $\{z_{t^\star}^{(r)}\}$ {within radius $r$}
    Select pilot subset $\{z_{t^\star}^{\mathrm{sub},(r)}\}$ of size $|B^{\mathrm{sub}}|$
    $\{s\}, \{z_0^{\mathrm{sub},(r)}\} \leftarrow \mathrm{Eval}(\{z_{t^\star}^{\mathrm{sub},(r)}\})$
    $\mathbf{S} \leftarrow \mathbf{S} \cup \{z_0^{\mathrm{sub},(r)} \mid s > \tau\}$
    Define remaining high-score batch at $t^\star$: $\{z_{t^\star}^{\mathrm{rest},(r)}\} \leftarrow \{z_{t^\star}^{(r)} \mid s > \tau\} \setminus \{z_{t^\star}^{\mathrm{sub},(r)}\}$
    **for** $t = t^\star, t^\star - 1, \ldots, 1$ **do**
        Sample $\{\boldsymbol{\varepsilon}\} = \{(\varepsilon_x, \varepsilon_h)\} \sim \mathcal{N}(\mathbf{0}, \boldsymbol{I})$
        Subtract center of mass from $\varepsilon_x$
        $\{z_{t-1}^{\mathrm{rest},(r)}\} \leftarrow \frac{1}{\sqrt{1-\beta_t}}\Big(\{z_t^{\mathrm{rest},(r)}\} - \frac{\beta_t}{\sqrt{1-\bar{\alpha}_t}}\,\varepsilon_\theta(\{z_t^{\mathrm{rest},(r)}\}, t)\Big) + \rho_t\{\boldsymbol{\varepsilon}\}$
    **end for**
    $\mathbf{S} \leftarrow \mathbf{S} \cup \{z_0^{\mathrm{rest},(r)}\}$
**end while**

---

Table 6: Wall-clock end-to-end consumption of different sampling strategies for generating 10,000 molecules on QM9. The experimental setting is identical to Table 2, using a fixed batch size of 100 and an intermediate timestep $t = 300$. 'Count' indicates how many attempts are required to produce 10,000 molecules (e.g., running inference with batch size 100 for 100 iterations). For models without DIST, the stages 'Diffuse' and 'Steer' correspond to the standard inference procedures $T \to t$ and $t \to 0$, respectively. The 'Total' column is reported in minutes and seconds (`mm:ss.ss`). The better strategy and its total wall-clock consumption are highlighted in **bold**.

| Model | Diffuse | | Duplication | | Pilot Evaluation | | Steer | | Total $\downarrow$ |
|---|---|---|---|---|---|---|---|---|---|
| | Time (s) | Count | Time (s) | Count | Time (s) | Count | Time (s) | Count | |
| EDM | 60.935 | 100 | – | – | – | – | 26.115 | 100 | 145:07.11 |
| **EDM+DIST** | 60.933 | 3 | 0.0009 | 258 | 27.0525 | 129 | 26.115 | 50 | **82:59.45** |
| GeoLDM | 58.1163 | 100 | – | – | – | – | 25.0704 | 100 | 138:39.72 |
| **GeoLDM+DIST** | 58.1163 | 3 | 0.0008 | 220 | 25.9830 | 110 | 25.0705 | 50 | **71:26.91** |
| RADM | 26.4737 | 100 | – | – | – | – | 11.3250 | 100 | 62:60.53 |
| **RADM+DIST** | 26.4742 | 3 | 0.0008 | 219 | 11.9224 | 66 | 11.4118 | 70 | **27:47.10** |

single hyperparameter of DIST while keeping all other settings fixed to the configuration of DIST strengthened EDM Hoogeboom et al. (2022) as described in Sec. 4. This ensures that every sub-experiment isolates the effect of one hyperparameter change.

As discussed in Sec. 3.2, DIST operates purely as an inference-time corrective module and does not require retraining or modification of the backbone diffusion model. Because inference is substantially cheaper than training (e.g., in DDPM (Ho et al., 2020), training takes 10.6 hours on 8 V100 GPUs while generating 256 CIFAR10 samples takes only 17 seconds), **hyperparameter search is lightweight in practice**. Importantly, the results below show that DIST performs **consistently well across a broad range of hyperparameters**, indicating that the method is not overly sensitive to tuning.

## H.1 BATCH SCORE THRESHOLD

Given a threshold $\tau$, DIST selects batches according to whether their pilot scores exceed this value, which determines the coverage of both the true distribution and the model distribution. Larger thresholds restrict selection to batches that are more likely to correspond to valid high-density regions, whereas smaller thresholds broaden the selected region and capture more mass, while at the cost of admitting samples that deviate from the true distribution (see Sec. 3.2).

The results are reported in Table 7. Although a larger $\tau$ improves stability metrics, the admissible region becomes overly constrained and may harm overall performance. Moreover, the retained ratio $r_c$ (see Appendix G.1) drops from 71% to 32%, indicating that a larger portion of batches is discarded, which in turn reduces the sampling efficiency. Across all tested values, DIST consistently outperforms the EDM (Hoogeboom et al., 2022), demonstrating robustness to the choice of $\tau$.

Table 7: Ablation on batch score threshold $\tau$ for molecule stability on QM9.

| $\tau$ | Atom Sta (%) | Mol Sta (%) | Valid (%) | Valid$\times$Unique (%) |
|---|---|---|---|---|
| EDM | 98.7 | 82.0 | 91.9 | 90.7 |
| 0.82 | 98.9 | 87.8 | 95.4 | 92.4 |
| 0.84 | 99.1 | 88.2 | 96.6 | 93.9 |
| 0.86 | 99.2 | 89.9 | 96.9 | 94.1 |
| 0.88 | 99.2 | 90.2 | 96.8 | 93.2 |

## H.2 INTERMEDIATE TIMESTEP

The intermediate timestep $t$ determines when the corrective selection is applied. If $t$ is too large, forward noise destroys the DC-structure; if too small, the candidate pool does not adequately represent the intermediate distribution (see Sec. 3.2).

As shown in Table 8, smaller $t$ values correct the distribution later in the reverse chain, improving stability metrics, while excessively small values may reduce uniqueness due to extended diffusing. Even without intentional tuning, tested $t$ values yield strong improvements over the EDM baseline, indicating that DIST remains effective across a broad range.

Table 8: Ablation on intermediate timestep $t$ on QM9.

| $t$ | Atom Sta (%) | Mol Sta (%) | Valid (%) | Valid$\times$Unique (%) |
|---|---|---|---|---|
| EDM | 98.7 | 82.0 | 91.9 | 90.7 |
| 200 | 99.2 | 90.2 | 96.8 | 92.5 |
| 300 | 99.2 | 89.9 | 96.9 | 94.1 |
| 400 | 99.2 | 89.7 | 96.0 | 94.0 |
| 500 | 99.1 | 89.3 | 95.4 | 92.1 |

## H.3 PERTURBATION INTENSITY

Following Definition 3.1 and the discussion in Sec. 3.2, the space is partitioned into radius-$r$ batches $\{B_j\}_{j=1}^J$, each serving as a local region around peaks. After obtaining the candidate pool at timestep $t$, these batches are optionally perturbed with Gaussian noise of scale $\lambda$.

Table 9 shows that DIST is highly robust to a small perturbation intensity: tested $\lambda$ values produce strong performance, and even $\lambda = 0$ (pure duplication without perturbation) already offers substantial improvements. This indicates that DIST does not rely on fine-grained tuning of perturbation intensity.

Table 9: Ablation on perturbation intensity $\lambda$ on QM9.

| $\lambda$ | Atom Sta (%) | Mol Sta (%) | Valid (%) | Valid$\times$Unique (%) |
|---|---|---|---|---|
| EDM | 98.7 | 82.0 | 91.9 | 90.7 |
| 0 | 99.2 | 89.9 | 96.9 | 94.1 |
| 0.05 | 99.2 | 89.7 | 95.9 | 93.1 |
| 0.1 | 99.1 | 90.1 | 95.9 | 92.8 |
| 0.2 | 99.3 | 90.3 | 96.2 | 92.7 |

