# OpenReview forum: "Diffuse and Steer: Corrective Sampling for Stable 3D Molecular Generation"
_ICLR.cc/2026/Conference — Submitted to ICLR 2026_

### Official Review · Reviewer_5oDq · 2025-10-27

**Soundness:** 3
**Presentation:** 3
**Contribution:** 2
**Rating:** 4
**Confidence:** 3

**Summary:**

This paper addresses the instability of diffusion models in 3D molecular generation. The authors identify that molecular data exhibits a dense-concentrated (DC) structure—valid molecules correspond to narrow, isolated peaks in distribution space separated by low-density regions. Standard diffusion models often drift into invalid regions due to this concentration, leading to unstable or chemically invalid molecules. To address this, the authors propose DIST, a plug-in corrective sampling module. DIST selectively corrects intermediate distributions during reverse diffusion by evaluating sample batches, discarding off-distribution trajectories, and steering remaining samples back toward high-density, valid regions. They provide theoretical guarantees and show that DIST improves molecular validity and stability across several backbone models on QM9 and GEOM-Drugs datasets, while cutting inference timesteps nearly in half.

**Strengths:**

This paper is well motivated and has some theoretical insights. The empirical results show consistent improvement across diverse backbones and datasets.

**Weaknesses:**

1. While the application to 3D molecules is new, the idea of filtering intermediate states resembles rejection or guidance sampling.
2. No comparison to alternative corrective techniques (e.g., score rescaling, gradient correction, or classifier guidance).
3. Scalability to very large molecules or protein-level systems is not demonstrated.
4. Dependence on heuristic thresholds (τ) and pilot sampling design may reduce reproducibility or generality.

**Questions:**

1. How sensitive is DIST’s performance to the choice of threshold and batch radius?
2. Is the theoretical bound (Proposition 3.1) empirically validated—e.g., by measuring total variation between qt and pt estimates?

---

> ### Author Response · Authors · 2025-11-20
> **Response to Reviewer 5oDq - Part I**
>
> We appreciate the insightful comments from the reviewer. We have revised the paper accordingly and provide the responses as below.
> ### Weaknesses
> - W 1: While the application to 3D molecules is new, the idea of filtering intermediate states resembles rejection or guidance sampling.
>   - The general inspiration of DIST aligns with this general paradigm of rejection sampling; **however, DIST introduces several key innovations:**
>       - It is the first corrective module tailored specifically for **3D molecular diffusion**, where the data distribution exhibits a unique dense-concentrated (DC) structure;
>       - It provides a **solid theoretical connection** between DC-structure, distributional contraction, and sampling drift; and
>       - It achieves **efficiency gains**, whereas many rejection or guidance methods incur significant additional computational cost due to repeated discarding or multiple forward passes.
>   For completeness, **we provide a detailed comparison between DIST and conceptually related works in Appendix B in the revision**.
>   - **We respectfully think this should not be a weakness of our work.** First of all, several key innovations have to be introduced as discussed above. Secondly,  rejection sampling and guidance-based sampling are well-established principles originating from statistics [1,2], and they have been applied widely across successful applications, including recent LLM token-generation methods [3] and multimodal diffusion models [4,5]. **By the same reasoning, these works would also be deemed unoriginal, which we believe is not a fair assessment.**
>
>
>
>
> - W 2: No comparison to alternative corrective techniques (e.g., score rescaling, gradient correction, or classifier guidance).
>   - To the best of our knowledge, DIST is the first *plug-in corrective module* designed specifically for **molecular diffusion generation**, and molecular data shows unique structural characteristics.  Alternative corrective techniques have been very successful in image generation, but they are **not directly applicable** to molecular data for two main reasons:
>     - **Most of these methods require an additional network or training objective or even inference timestep cost**, which introduces substantial computational overhead and compromises efficiency. This contradicts our motivation of designing a lightweight plug-in module that improves performance *without* increasing training or sampling cost.
>     - **Molecular data have fundamentally different structure from images.** Their distribution exhibits a highly concentrated geometry (DC-structure), and the gradients used in guidance or score correction do not reliably correspond to chemically valid directions in 3D space. As a result, image-friendly corrective techniques do not transfer well to molecular settings.
>
>   - **We have added new explanations and experiments in Sec. 3.1 and Appendix D in the revised paper**, demonstrating that even the widely adopted corrective technique such as DDIM [6], which often improves image quality but actually degrades molecular stability compared to DDPM [7]. **This empirical observation further supports our claim that molecular data require a different form of corrective mechanism, and motivates the design of DIST as a principled and efficient solution for molecular generation.**
>
>
> - W 3: Scalability to very large molecules or protein-level systems is not demonstrated.
>   - We would like to clarify that **small-molecule generation and protein-level generation are fundamentally different tasks**. Throughout this work, the term *molecule* (or *molecular data*) refers specifically to **small organic molecules**; when referring to macromolecules, we explicitly use *peptides* or *proteins* to avoid ambiguity.
>
>     Importantly, the **data distributions** of small molecules and proteins differ substantially. Small-molecule configurations exhibit a **highly concentrated and discontinuous** structure:
>     - Small perturbations to atomic positions can easily break valency or violate chemical rules, rendering the structure **invalid**;
>     - Conversely, small but precise changes can produce **distinct valid molecules**, reflecting strong **mode concentration**.
>
>     In contrast, **protein conformations are much smoother**:
>     - Local fluctuations in side chains typically do not change the global fold;
>     - Geometric noise is averaged out rather than producing a new protein;
>     - The probability landscape is therefore **less concentrated** and more tolerant to perturbations.
>
>     From a practical perspective, **molecular generation and protein modeling do not share the same models, datasets, or evaluation metrics. Thus, scalability to protein-level systems lies outside the intended scope of this work.** Nevertheless, we thank the reviewer for raising this point, and we have added corresponding statements to the future work section (Sec. 5) in the revised manuscript.

---

> ### Author Response · Authors · 2025-11-20
> **Response to Reviewer 5oDq - Part II**
>
> - W 4: Dependence on heuristic thresholds (τ) and pilot sampling design may reduce reproducibility or generality.
>   - **We provide additional ablation studies and discussion in Appendix H** of the revised paper. These results show that DIST maintains **strong and stable performance across a wide range of hyperparameter choices**, indicating that its behavior is **not sensitive to the specific value of the threshold $\tau$ or to details of the pilot sampling design.** **The pilot sampling component is introduced primarily for efficiency rather than for controlling performance.**
>   - Moreover, DIST is a **lightweight plug-in module**, making it **flexible and easy to adapt** without compromising reproducibility.
>
> ### Questions
> - Q 1: How sensitive is DIST’s performance to the choice of threshold and batch radius?
>   - **We provide additional ablation studies in Appendix H** of the revised paper. These experiments vary the threshold and the perturbation intensity (within radius-$r$), and the results show that DIST maintains stable performance across a wide range of settings. This demonstrates that DIST is robust and not sensitive to the specific choice of these hyperparameters in practice. For example, when combined with EDM, **DIST improves molecule stability by more than 5\%** across thresholds from 0.82 to 0.88 and perturbation scales from 0 to 0.1.
> - Q 2: Is the theoretical bound (Proposition 3.1) empirically validated—e.g., by measuring total variation between qt and pt estimates?
>   - Proposition 3.1 establishes that **DIST corrects $q_t$ toward the true reverse distribution $p_t$** by matching the model coverage $\beta$ with the true coverage $\alpha$ via selective correction. When the generative performance is satisfied, the empirical distribution of samples after correction implicitly confirms that $q_t$ remains close to $p_t$.
>
>   - As with all diffusion-based generative models, directly computing the total variation distance between $q_t$ and $p_t$ is intractable, since neither distribution admits a closed-form density in high-dimensional molecular spaces. **Consequently, convergence between $q_t$ and $p_t$ can only be assessed indirectly through sample quality, validity, and stability—this is the standard practice across diffusion literature.**
>
>   - As noted in the original submission, the deviation can still be **approximated using sample-based estimators**. For the reviewer’s interest, **the revised version provides an explicit discussion of this approximation and the associated confidence bound in Appendix E.3**, further supporting the empirical consistency of Proposition 3.1.
>
> [1] Gordon et al., Novel Approach to Nonlinear/Non-Gaussian Bayesian State Estimation, 1993.
>
> [2] von Neumann, Various Techniques Used in Connection with Random Digits, 1951.
>
> [3] EAGLE: Speculative Sampling Requires Rethinking Feature Uncertainty, 2024.
>
> [4] GLIDE: Photorealistic Image Generation and Editing with Text-Guided Diffusion Models, 2021.
>
> [5] Dhariwal & Nichol, Diffusion Models Beat GANs on Image Synthesis, 2021.
>
> [6] Denoising Diffusion Implicit Models, ICLR 2021
>
> [7] Denoising Diffusion Probabilistic Models, NeurIPS 2020

---

> > ### Author Response · Authors · 2025-11-28
> > **Follow-up on Our Responses - Reviewer 5oDq**
> >
> > Dear Reviewer 5oDq,
> >
> > Thank you once again for the time and care you devoted to reviewing our work. Since more than a week has passed since we submitted our rebuttal, we wanted to check in and make sure that all of your questions have been fully resolved.
> >
> > - If any parts of our response remain unclear, or if further explanation would be helpful, we would be very glad to provide additional details.
> > - In our revised manuscript and rebuttal, we have made several substantive additions, including:
> >   - **Comprehensive ablation studies** that examine the sensitivity of DIST and confirm its robustness,
> >   - **New experiments** that directly demonstrate the presence and influence of DC-structure,
> >   - **An expanded related-work discussion** that clarifies the novelty and broader significance of DIST.
> >
> > We believe that the main issues you raised have now been thoroughly addressed, and that some earlier concerns stemmed from misunderstandings that we have worked carefully to clarify.
> >
> > If you feel that the revised version resolves your concerns, we would sincerely appreciate your reconsideration of the rating.
> >
> > Thank you again for your constructive feedback and for engaging deeply with our work.
> >
> > Warm regards,
> > **The Authors**

---

### Official Review · Reviewer_wDAK · 2025-10-31

**Soundness:** 2
**Presentation:** 3
**Contribution:** 2
**Rating:** 2
**Confidence:** 3

**Summary:**

This work focuses on the "dense-concentrated (DC) structure" inherent to molecular data: chemically valid structures form sharp, densely packed peaks in the representation space, separated by regions of near-zero density. This structure makes diffusion modeling fragile. To mitigate this, this paper propose DIST (DIffuse and STeer), a selective correction method. DIST filters and rescales intermediate distributions during inference, steering trajectories toward the valid molecular peaks. Experimental results demonstrate that DIST consistently enhances performance.

**Strengths:**

1. The identification of the "dense-concentrated structure" in molecular data as a critical challenge is valuable.

2. The theoretical analysis enhances the plausibility of the proposed approach.

**Weaknesses:**

1. A thorough survey and comparison to the literature on "exposure bias in diffusion models" (such as [1], [2], [3], and recent works)  are highly necessary. The proposed solution shares conceptual similarities with existing methods—for example, Proposition 2 in [2] reaches conclusions analogous to this work’s Cor. 3.1 and Prop. 3.1, albeit with different metrics. The absence of a clear comparison from existing methods makes it difficult to assess the novelty and uniqueness of its contribution.

2. Lack of experimental validation for the issue of dense-concentrated data: The phenomenon observed in Table 1 mirrors exposure bias and which is also found in image generation with diffusion models. To substantiate the arguments in Section 3.1, additional experiments should be conducted to demonstrate that dense-concentrated data exhibit more pronounced exposure bias than smoother distributions.

3. Clarity and rigor in writing need improvement:
    - Line 230: The derivation of \(\|\nabla \log p(z_t)\|\) and Equations 6, 7 needs rigorous mathematical justification.
    - Section 3.1 claims that the DC-structure causes the "reverse update to step past the peak and cross into high-density opposite regions." I suggest including a 1D Gaussian mixture simulation example, illustrating how specific choices of \(\sigma_*\), \(\Delta\), and \(m_k\) lead to sampling falling into low-density regions.
    - I suggest including a pseudocode implementation of the DIST algorithm for reproducibility.
    - Minor typo: Line 306 "the the reverse process".

**Questions:**

1. The DIST algorithm involves multiple manually tuned hyperparameters (intermediate timestep t, filtering threshold, batch division, pilot sample ratio). What guidelines exist for tuning them? If time permits, an ablation study for each parameter is recommended.

2. Intuitively, transforming the data (e.g., stretching the "x-axis") could reshape the distribution to be sparser and smoother, eliminating the dense-concentration. Is the DC-structure an inherent property of molecular data, or an artifact of representation? A discussion on this would strengthen the work’s motivation.

---

> ### Author Response · Authors · 2025-11-20
> **Response to Reviewer wDAK - Part I**
>
> Thank you for the insightful suggestions and we have revised the paper. Our point-to-point responses are as below.
> ### Weaknesses
> - W 1: A thorough survey and comparison to the literature on "exposure bias in diffusion models" (such as [1], [2], [3], and recent works) are highly necessary. The proposed solution shares conceptual similarities with existing methods—for example, Proposition 2 in [2] reaches conclusions analogous to this work’s Cor. 3.1 and Prop. 3.1, albeit with different metrics. The absence of a clear comparison from existing methods makes it difficult to assess the novelty and uniqueness of its contribution.
>   - We thank the reviewer for raising this point. **The references labeled as [1], [2], and [3] in the review are not given**, so we are currently **unable to identify which works the reviewer is referring to**. Could the reviewer kindly clarify these citations? **We will begin working on a detailed comparison immediately once the intended references are clarified.**
> - W 2: Lack of experimental validation for the issue of dense-concentrated data: The phenomenon observed in Table 1 mirrors exposure bias and which is also found in image generation with diffusion models. To substantiate the arguments in Section 3.1, additional experiments should be conducted to demonstrate that dense-concentrated data exhibit more pronounced exposure bias than smoother distributions.
>   - This is an interesting perspective. We agree that additional experimental evidence comparing dense-concentrated molecular data with smoother distributions will improve the rigour of the manuscript. In the revised version, **we provide further explanation and new experiments in Sec.3.1 and Appendix D**, showing that molecular data with DC-structure behaves **fundamentally differently under diffusion sampling and indeed exhibits significantly stronger exposure bias than smoother distributions.**
> - W 3:
>   1. Line 230: The derivation of (|\nabla \log p(z_t)|) and Equations 6, 7 needs rigorous mathematical justification.
>   - **We provide the math justification in Appendix C** for the derivation of Equations 6 and 7 in the revised paper.
>   2. Section 3.1 claims that the DC-structure causes the "reverse update to step past the peak and cross into high-density opposite regions." I suggest including a 1D Gaussian mixture simulation example, illustrating how specific choices of (\sigma_*), (\Delta), and (m_k) lead to sampling falling into low-density regions.
>   - **We provide both 2D and 1D toy examples to directly illustrate the phenomena caused by DC-structure and the overshoot mechanism**. These Mixture-of-Gaussian examples and the code to generate them are included in the anonymous repository (https://anonymous.4open.science/r/Mixture-of-Gaussian-Toy-Example-3A24/README.md), offering a clean, semantics-free visualization of how DC-structure lead to overshoot and drift in diffusion sampling.
>   3. I suggest including a pseudocode implementation of the DIST algorithm for reproducibility.
>   - **The pseudocode implementation of the DIST algorithm is provided in Appendix G.2** in the revised paper.
>   4. Minor typo: Line 306 "the the reverse process".
>   - We thank the reviewer for the careful check and it has been fixed in the revision.

---

> > ### Comment · Reviewer_wDAK · 2025-11-22
> > **Supplementing citations for W1**
> >
> > I am sorry for missing the citations for the first weakness. They are provided as follows:
> > [1] Elucidating the Exposure Bias in Diffusion Models, ICLR2024, https://arxiv.org/abs/2308.15321
> > [2] Improved Diffusion-based Generative Model with Better Adversarial Robustness, ICLR2025, https://arxiv.org/abs/2502.17099
> > [3] Alleviating Exposure Bias in Diffusion Models through Sampling with Shifted Time Steps, ICLR2024, https://arxiv.org/abs/2305.15583

---

> > > ### Author Response · Authors · 2025-11-22
> > > **Response to Reviewer wDAK - Regarding the Three References**
> > >
> > > ### Weaknesses
> > > - W 1: A thorough survey and comparison to the literature on "exposure bias in diffusion models" (such as [1], [2], [3], and recent works) are highly necessary. The proposed solution shares conceptual similarities with existing methods—for example, Proposition 2 in [2] reaches conclusions analogous to this work’s Cor. 3.1 and Prop. 3.1, albeit with different metrics. The absence of a clear comparison from existing methods makes it difficult to assess the novelty and uniqueness of its contribution.
> > >
> > >   - We thank the reviewer for letting us know the citations and highlighting the connection to exposure bias issue. We include the discussion in **Appendix B** of the revision, and provide the response below. After examining the three referenced works [1,2,3], we believe DIST is fundamentally different from them in motivation, theory, and methodology:
> > >     -  **Conceptual motivation.**
> > >      The exposure-bias works [1,2,3] treat the problem as a general *training–inference mismatch*: **they analyze how prediction errors accumulate across diffusion steps and how to make adjacent transitions more stable**. In contrast, **DIST is from the data distribution perspective**. We formalize structure of **molecular distributions**, which is different from smoother distributions like image data distribution, and identify it as the source of the severe fragility in molecular diffusion. This perspective does not appear in any of [1,2,3].
> > >     -  **Theoretical objective.**
> > >      The three works study local transition behavior:  [1] analyzes how prediction error increases the variance of the reverse transition; [2] shows that training loss is well-behaved only when the sample stays inside a KL-ball around training data; [3] identifies better-coupled timesteps to reduce mismatch in a single transition. **These analyses all concern step-wise error propagation.** In contrast, **DIST provides a guarantee on the final distribution**: Cor. 3.1 and Prop. 3.1 show that correcting the intermediate model distribution ensures closeness of the entire distribution at $t=0$ to the true one. Thus DIST addresses *global distributional correctness*, not only accuracy of a local transition.
> > >
> > >     -  **Practical method.**
> > >      All three exposure-bias methods introduce additional training or computation overhead: [1] modifies the variance schedule; [2] performs adversarial / DRO-style robust training; [3] searches for better timesteps based on predicted noise. DIST, in contrast, introduces a **training-free, plug-in selective correction module** that directly steers the *intermediate distribution* using pilot scoring. It is the only method among these that **offers NFE-level efficiency gains** (i.e., reductions in the number of *network function evaluations*, which in diffusion models correspond directly to the number of denoising timesteps) **while requiring no retraining of the backbone model** (i.e., the weights of the pretrained backbone model remain the same).
> > >
> > >
> > >   - Regarding Proposition 2 in [2]: it states that if the reverse transition from timestep $t+1$ to timestep $t$ is accurately approximated by the network, and if the initial distributions at $T$ are close, then the resulting model distribution and true distribution at $t$ remain close. This is a *local consistency* statement about one step. **DIST’s Cor. 3.1 and Prop. 3.1 address a different question:** if the *intermediate model distribution* is corrected to match the true one, then the *final distribution at $t=0$* is provably to be close. DIST also provides a mechanism called selective correction to enforce this condition.  **Therefore the two results are conceptually and mathematically distinct, with little direct overlap.**
> > >   - **In summary, DIST is original in its **intuition**, **theoretical formulation**, and **practical mechanism**. Unlike existing exposure-bias approaches, DIST is the only training-free plug-in method that achieves NFE-level efficiency (as we discuss in Appendix G of the revision), while also offering a theoretical guarantee on correcting the final distribution (as we discuss in Sec.3.2 in the original submission). We've included a discussion on the comparison with these three works in Appendix B in our revision. For completeness, we further include comparisons with additional, more directly relevant very recent works, going beyond the ones mentioned by the reviewer**.
> > >
> > >
> > >   [1] Elucidating the Exposure Bias in Diffusion Models, ICLR2024
> > >
> > >   [2] Improved Diffusion-based Generative Model with Better Adversarial Robustness, ICLR2025
> > >
> > >   [3] Alleviating Exposure Bias in Diffusion Models through Sampling with Shifted Time Steps, ICLR2024

---

> > > > ### Comment · Reviewer_wDAK · 2025-11-25
> > > >
> > > > Thank you for your detail response.
> > > >
> > > > The findings in Table 5 are quite interesting, as they clearly demonstrate a significant domain gap. The 2D and 1D toy examples are also intuitive and these could be included in the appendix of the paper.
> > > >
> > > > I have a few additional follow-up points:
> > > >
> > > > 1. Regarding the statement *“To the best of our knowledge, there is currently no existing corrective method that directly steers intermediate distributions in diffusion-based molecular generation”*: this wording may not be precise. Methods like guidance or temperature adjustment in molecular generation can already steer intermediate distributions to align with desired distributions. Perhaps what you intend to convey is that selective sampling strategies (of the type used in your work) are less common in molecular diffusion generation?
> > > >
> > > > 2. For the equation below Eq. 10: Could you clarify how this equation is derived? It appears to approximate Eq. 5 by assuming $\beta_t \to 0$ and ignoring the stochastic term—this approximation seems unreasonable. Even if we accept this approximation, there is a missing coefficient $-\frac{1}{\sqrt{1-\bar\alpha_t}}$ between the score function and the noise term. With this coefficient included, the equation $\|z_{t-1} - z_t\| \approx \beta_t \|\nabla \log p(z_t)\|$ would reduce to a trivial result (approaching 0), which undermines its meaning.

---

> ### Author Response · Authors · 2025-11-20
> **Response to Reviewer wDAK - Part II**
>
> ### Questions
> - Q 1: An ablation study for each hyperparameter is recommended.
>   - **An extensive ablation study has been added to Appendix H** in the revised paper, where we examine each hyperparameter individually and demonstrate that **DIST is robust across a wide range of settings**. **These results show that DIST requires minimal tuning in practice, and reasonable default choices consistently yield strong performance.** For example, when applied to EDM, DIST improves molecule stability by **at least 5\%** across a wide range of intermediate timesteps (200–500), perturbation intensities (0 to 0.1), and thresholds (0.82 to 0.88).
>
> - Q 2: Intuitively, transforming the data (e.g., stretching the "x-axis") could reshape the distribution to be sparser and smoother, eliminating the dense-concentration. Is the DC-structure an inherent property of molecular data, or an artifact of representation? A discussion on this would strengthen the work’s motivation.
>   - **There is a misunderstanding.** In fact, simple coordinate transformations (such as stretching or rescaling the spatial axis) do **not** alter the underlying DC-structure of molecular data. In diffusion models (or more generally in the machine learning field), the representation and learning spaces scale proportionally—if all coordinates are multiplied by a constant factor, both the input and the model output scale by the same factor, leaving the normalized data distribution unchanged.
>   - The DC-structure does not arise from numerical scaling or coordinate choice but is instead a direct consequence of **chemical and physical constraints** such as fixed bond lengths, angular rigidity, and discrete valency configurations. We have further clarified this point and provided empirical evidence in the response to **W 2 (Sec.3.1 and Appendix D)**, which highlight both the **specificity of molecular data** and the **existence of DC-structure** compared with smoother modalities such as images.

---

> ### Author Response · Authors · 2025-11-26
> **Second Round Responses to Reviewer wDAK**
>
> We thank the reviewer for the timely follow-up. We have revised the paper accordingly and provide the responses as below.
>
> - Suggestion: The 2D and 1D toy examples are also intuitive and these could be included in the appendix of the paper.
>
>   - The toy examples have been included in **Appendix C.2** of the revision.
>
> - Q 1: Regarding the statement “To the best of our knowledge, there is currently no existing corrective method that directly steers intermediate distributions in diffusion-based molecular generation”: this wording may not be precise. Methods like guidance or temperature adjustment in molecular generation can already steer intermediate distributions to align with desired distributions. Perhaps what you intend to convey is that selective sampling strategies (of the type used in your work) are less common in molecular diffusion generation?
>
>    - This is a very insightful comment. **There is a fundamental difference between DIST and methods such as guidance or temperature adjustment.**
>    - **TL;DR:**
>      - **Guidance requires additional information specifically designed for each task to modify the score function, whereas DIST does not require additional information. There are no existing works on guidance that "correct" the intermediate distribution for 3D molecular diffusion.**
>      - **Temperature scaling, on the other hand, is not corrective or even countercorrective by design (it reshapes the learned distribution toward high-density modes).**
>    - **Details:**
>      - Guidance alters the sampling process by modifying the sampling score function, $\nabla_x \log p$, using additional information to change the trajectory in a direction different from that given solely by the learned score. **However, DIST does not change the sampling score function, nor does it require additional prior knowledge to serve as the guidance.** **Moreover, guidance methods typically require task-specific designs.** While several such approaches have been proposed in other domains, such as image generation, **none exist for correcting the intermediate distribution of 3D molecular generation.**
>      - **Temperature adjustment is NOT corrective.** Temperature adjustment (variance scaling) is a technique that has been commonly adopted in protein diffusion/flow models [1, 2, 3]. It simply sharpens the sampled distribution towards high-density modes, which usually improve designability of proteins at the cost of diversity (see Sec. 3.4 in [1]). This could be an effective approach in protein generation where designability is highly important. In contrast, DIST does not steer toward only high-density samples but correct samples, which can be relatively low-density as well. **This approach does not steer the intermediate learned distributions to align with the desired ones; instead, it purposefully misaligns with the desired ones to reshape the learned distribution toward high-density modes.**
>
> - Q 2: For the equation below Eq. 10: Could you clarify how this equation is derived? It appears to approximate Eq. 5 by assuming and ignoring the stochastic term—this approximation seems unreasonable. Even if we accept this approximation, there is a missing coefficient between the score function and the noise term. The equation would reduce to a trivial result (approaching 0), which undermines its meaning.
>
>   - **There is a misunderstanding.** The equation below Eq.10 is not intended to approximate the full stochastic reverse process in Eq. 5. In order to analyze the overshoot mechanism of trajectories, we must **isolate the deterministic displacement of the reverse update**. The stochastic noise term is deliberately omitted, because **a Gaussian perturbation can move a trajectory to any location in the space and therefore obscures the geometric effect we aim to study**.
>   - To avoid confusion, we have rewritten the expression as $\,\||z_{t-1} - z_t\||_{\mathrm{det}}\,$ in the revision, and **we now provide a detailed derivation and justification of this deterministic approximation in Appendix C.3**. The purpose of this analysis is solely to **characterize the geometric drift (overshoot) induced by the DC-structure, not to approximate the full stochastic dynamics.**
>
> [1] Proteina: Scaling Flow-based Protein Structure Generative Models, ICLR 2025
>
> [2] SE(3)-Stochastic Flow Matching for Protein Backbone Generation, ICLR 2024
>
> [3] ReQFlow: Rectified Quaternion Flow for Efficient and High-Quality Protein Backbone Generation, ICML 2025

---

> > ### Comment · Reviewer_wDAK · 2025-11-28
> >
> > **For your reply to Q2:**
> > Thank you for adding details in Appendix C.3. However, I remain skeptical about **the correctness of the derivation** (I apologize for any confusion caused if I have not expressed this clearly before). Let me clarify in detail:
> >
> > **(a)** A critical coefficient that remains overlooked: There is a coefficient of
> > $-\frac{1}{\sqrt{1-\bar{\alpha}_t}}$
> >  between
> >
> > $s_\theta$
> >  (the network fitting $ \nabla\log p(z_t)$) and $\epsilon_\theta$ (the network fitting noise),
> > that is
> >
> > $s_\theta =-\frac{1}{\sqrt{1-\bar{\alpha}_t}}epsilon_\theta$ .
> >
> >
> > In your new derivation (line 930), you directly replace \( \epsilon_\theta \) with \( \nabla\log p(z_t) \), this  omits the key coefficient
> > \( -\frac{1}{\sqrt{1-\bar{\alpha}_t}} \)
> > (My main point here.)
> >
> > Besides it also requires a note that $\nabla\log p(z_t)$ is substituted for the network-predicted $s_\theta$ (under the assumption that the network is well-fitted).
> >
> > **(b)** To recap your new content (lines 898–931): I understand you aim to estimate $A_t = \frac{1}{\sqrt{1-\beta_t}} - 1$ and $B_t = \frac{\beta_t}{\sqrt{1-\beta_t}\sqrt{1-\bar{\alpha}_t}}$ when \( \beta_t \to 0 \), and you use "≈" (instead of formal limit language) for small \( \beta_t \).
> > You apply a Taylor expansion to get $A_t \approx 0$, factor$B_t$ as $\frac{\beta_t}{\sqrt{1-\bar{\alpha}_t}} \cdot \frac{1}{\sqrt{1-\beta_t}}$,
> > and approximate $\frac{1}{\sqrt{1-\beta_t}} \approx 1$.
> >
> > However, accounting for the coefficient in (a), the relevant coefficient to estimate is
> > $\tilde{B} := \frac{\beta_t}{\sqrt{1-\beta_t}}$
> > (since \( \frac{1}{\sqrt{1-\bar{\alpha}_t}} \)
> > is absorbed into the score function).
> >
> > Using limit notation simplifies this:
> >  $\lim_{\beta_t \to 0}$
> > $A_t = 0$
> > and
> > $\lim_{\beta_t \to 0}$
> >  $\tilde{B} = 0$.
> > This leads to the "trivial (vanishing) result" I mentioned earlier.
> >
> > I hope this is clearer now.
> >
> > **For your reply to Q1:**
> > Thank you for your prompt and helpful reply.  To clarify my earlier comment: I fully understand the fundamental differences in their mechanisms and goals. My only concern was that the original statement’s wording felt somewhat broad, which might lead readers (like myself) to interpret "directly steers intermediate distributions" as simply altering the original diffusion model’s intermediate distributions—an interpretation that would inadvertently include those existing methods. This is not a critique of your work’s novelty, but rather a suggestion to refine the phrasing for greater precision.

---

> > > ### Author Response · Authors · 2025-11-29
> > > **Third Round Responses to Reviewer wDAK**
> > >
> > > We thank the reviewer for the feedback. We have revised the paper accordingly and provide the responses as below.
> > >
> > > - Q 1: the original statement’s wording felt somewhat broad, which might lead readers (like myself) to interpret "directly steers intermediate distributions" as simply altering the original diffusion model’s intermediate distributions. This is not a critique of your work’s novelty, but rather a suggestion to refine the phrasing for greater precision.
> > >   - We thank the reviewer for this helpful suggestion. Our intention was to emphasize that DIST **directly** steers the intermediate **distribution**, whereas existing approaches in molecular diffusion generation typically influence sampling only in an indirect or implicit manner (e.g., through per-step score adjustments). We agree that the original wording could be interpreted more broadly than intended, especially for readers less familiar with the distinction between step-wise trajectory modifications and distribution-level correction. To improve clarity, we have refined the phrasing in the revision and italicized the word “directly” to highlight this distinction more clearly.
> > > - Q 2: considering the coefficient in $s_\theta=$  $-\frac{1}{\sqrt{1-\bar{\alpha}_t}} \epsilon\_\theta$, using the limit notation simplifies $A_t$ and $\tilde{B}_t$ may lead to the "trivial (vanishing) result", as $\frac{1}{\sqrt{1-\bar{\alpha}_t}}$ can be absorbed into the score function.
> > >   - **This appears to be a misunderstanding, and we now provide a detailed explanation and derivation in Appendix C.3 of the revision.** The coefficient $\frac{1}{\sqrt{1-\bar{\alpha}\_t}}$ cannot be absorbed into the score function in our analysis. In practice, this term **diverges** as $t \to 0$, and therefore cannot be treated as a stable or negligible factor when characterizing the deterministic displacement. Moreover, it is the network output $\epsilon_\theta$, rather than $s_\theta$, that operates on the same scale as $z_t$; thus this coefficient plays an essential role and cannot be omitted. The limit notation is introduced solely to simplify the deterministic expression of the displacement, not to collapse the term into a vanishing quantity.

---

### Official Review · Reviewer_DXTg · 2025-10-31

**Soundness:** 3
**Presentation:** 2
**Contribution:** 3
**Rating:** 4
**Confidence:** 3

**Summary:**

This paper focuses on the vulnerability of diffusion models in 3D molecular generation caused by intermediate-time drift and extremely narrow effective domain. The authors first formally propose the data distribution hypothesis of "dense-centralized (DC) structure," and based on this, analyze the overshoot problem in back-inference (Equations (6)(7)). They then propose a pluggable bias-correcting sampling framework, DIST (Diffuse and Steer): at intermediate time points, replication-perturbation forms local "batches," which are scored using pilot inference and inconsistent/low-quality batches are filtered out to obtain the corrected intermediate distribution q_{t}^{c} before continuing back-sampling. Empirical results on QM9 and GEOM-Drugs show improved stability and legitimacy for various backbones (EDM/GeoLDM/RADM), and claim a nearly 50% reduction in the average number of steps.

**Strengths:**

1.Abstracting the molecular distribution into a DC structure of "multiple narrow peaks + low-density spacing" provides a theoretical explanation and intuitive illustration of the trajectories intersecting in the intermediate distribution and the misleading effects of the mean-based score field.

2.Without altering the backbone weights and hyperparameters, the algorithm directly selects the best and discards the worst in the inference process, empirically demonstrating stable performance improvements on multi-backbone architectures and two datasets.

3.The author provides a step count calculation and shows in a table the significant decrease in average steps.

4.Cor. 3.1 gives the conclusion that "closer intermediate distributions lead to closer final distributions" as the TV distance does not increase; Prop. 3.1 gives the upper bound of the error after selective correction as dependent on α(τ), β(τ) and the conditional TV deviation.

**Weaknesses:**

1.Cor. 3.1 actually utilizes the general fact that "any Markov kernel is 1-Lipschitz (non-expansion) on TV" (k∈[0,1] often only takes the value 1), but does not guarantee strict contraction (k<1). Calling it "TV–contraction" is misleading to the reader into thinking it means "inevitable contraction." It is suggested to change it to "non-expansive step / TV non-expansion," and clarify the typicality of k=1.

2.The upper bound of Prop. 3.1 depends on the quality of the true distribution, but in practice the true distribution is unavailable. The authors approximate it with "pilot inference results (stability/legitimacy)"; however, the consistency of this approximation and the confidence bound (the estimation error of α, β) are not analyzed, so the upper bound is difficult to instantiate.

3.The paper calculates the average number of steps (e.g., (T−t)/|B|+t) based on "replicating batches up to t + full replay of a small number of pilot samples + continuing only for batches that pass the threshold." However, the description of the complete reverse cost of the pilot samples is somewhat vague. The "pilot cost of discarded batches" should also be included in the total budget before comparing the end-to-end total cost with the baseline's fixed 1000 steps. It is recommended to provide the precise accounting formula and pseudocode in Appendix D, and report the wall-clock time with the same hardware and parallelism.

4.Only rule-based indicators such as stability, legitimacy, and uniqueness are reported. It is recommended to add: skeleton diversity , ring structure and heteroatom distribution, QED/SA/synthetic feasibility, chirality and geometric configuration preservation, etc., to reflect the impact of the "screening + replication" strategy on diversity and pattern coverage.

5.The settings and scales of "radius r", the trade-off between quality and diversity in "replication count/perturbation intensity", the specific definition and threshold selection of pilot score s_{j} (which seems to be based on stability/legitimacy in the final draft), and the sensitivity to random seed/batch size are currently less detailed in the main text.

**Questions:**

1.The abstract claims that "computational overhead is reduced to nearly half that of the standard number of steps." Please provide the total end-to-end time (including pilot cost and the overhead of dropped batches) and the equivalent GPU hours, and compare it with the baseline on the same hardware and concurrency level.

2.The image comparison in Fig. 1 is intuitive, but could the differences be exaggerated due to variations in image task semantics, etc.? Could you provide a 1D/2D toy density comparison to more purely represent the mechanism of "multiple narrow peaks + overlap + overshoot"?

3.The reverse update of Equation (5) uses ϵ_{θ} to approximate the score; in molecular tasks, the SE(3) isovariant network or coordinate alignment is often used to reduce ambiguity. The authors' subsequent experiments included isovariant and non-isovariant backbones, but did they examine the sensitivity differences of DIST under canonicalization/coordinate normalization schemes?

4.Please change "TV–contraction" to "TV non-expansion" and clarify that κ=1 in general. Furthermore, providing sufficient conditions under which κ<1 would be more convincing.

5.Please provide accurate accounting (including pilot costs for failed batches) in Appendix D, and supplement with wall-clock and throughput to avoid substituting actual time with just "steps".

---

> ### Author Response · Authors · 2025-11-20
> **Response to Reviewer DXTg - Part I**
>
> We thank the reviewer for the valuable suggestions for improvement and have revised our paper. Our responses to the reviewer’s comments are below:
> ### Weaknesses
> - W 1:  Change Cor. 3.1 to "non-expansive step / TV non-expansion," and clarify the typicality of k=1.
>   - **There is a misunderstanding** regarding the general non-expansiveness of Markov kernels in TV distance $$
> \|K_{t\to0}q_t - K_{t\to0}p_t\|_{TV} \leq \kappa\|q_t - p_t\|\_{TV}, \quad \kappa \in [0,1].$$
>
>     While this is true in full generality, the situation in our diffusion setting is more specific. The reverse transition $K_{t\to 0}$ is a *continuous Gaussian* kernel, which necessarily induces overlap between transition densities and rules out cases of mutual singularity. This structural property yields a **strict inequality $\kappa < 1$** for the kernels used in our model, rather than the borderline case $\kappa = 1$ that may occur in arbitrary Markov systems.
>   - **For clarity, in Appendix E.1 in our revision, we have included an argument establishing this stricter bound for Gaussian transitions marked in red.** Throughout the paper, the term *TV-contraction* is used in this setting precisely.
>
>
>
>
> - W 2: The consistency of Prop. 3.1 approximation and the confidence bound (the estimation error of α, β) are not analyzed, so the upper bound is difficult to instantiate.
>   - The theoretical bound in Prop. 3.1 involves the true coverage terms  $$
> \alpha(\tau)=\sum_{j\in J^\star(\tau)}\pi_j,
> \qquad
> \beta(\tau)=\sum_{j\in J^\star(\tau)}\hat{\pi}_j,
> $$which represent the portions of true and model probability mass preserved by selective correction. In practice, **these quantities can be empirically estimated** from available samples of both the forward and reverse diffusion processes, and their estimation errors can be rigorously bounded. **A thorough estimation is provided in Appendix E.3 in the revised paper.**
>   - We would also like to clarify that both coverage terms $\alpha(\tau)$ and $\beta(\tau)$ are **functions of the selection threshold** $\tau$. As stated in the paper (Section 3.2, before Eq. (9)):
>     > 'Smaller thresholds $\tau$ restrict the selection to batches that are more likely to correspond to valid regions, reducing coverage; larger thresholds broaden the selection and capture more mass, but at the cost of admitting regions inconsistent with the true distribution.'
>
>     We have **included an ablation study on the threshold selection in Appendix H.1 in the revised paper**. The ablation study empirically confirms how varying $\tau$ affects the performance of the selective correction. **The results complement the theoretical bound by showing that the proposed method remains stable across a range of reasonable thresholds.**
>
> - W 3: The "pilot cost of discarded batches" should also be included in the total budget. It is recommended to provide the precise accounting formula and pseudocode, and report the wall-time with the same hardware and parallelism.
>   - **There is a misunderstanding regarding the pilot cost.** In the original submission, the pilot cost of discarded batches had been included in all of our efficiency analyses, including the **results in Table 3** and the **total cost formula and illustrative example in Appendix D** in the original submission (note that in the revised version, this material appears in Appendix G.1 as we've inserted additional sections).
>   - We have also **provided pseudocode for the cost computation and to report wall-clock time** measured under identical hardware and parallelism settings in **Appendix G.2** in our revision, as recommended. For example, **augmenting EDM with DIST reduces the end-to-end sampling time from 145 minutes to 83 minutes**, which aligns precisely with the corresponding reduction in effective timesteps (from 1000 to 556) in Table 3.

---

> ### Author Response · Authors · 2025-11-20
> **Response to Reviewer DXTg - Part II**
>
> - W 4: Only rule-based indicators such as stability, legitimacy, and uniqueness are reported. It is recommended to add: skeleton diversity , ring structure and heteroatom distribution, QED/SA/synthetic feasibility, chirality and geometric configuration preservation, etc., to reflect the impact of the "screening + replication" strategy on diversity and pattern coverage.
>   - **The requested metrics are not applicable in our setting. DIST exactly follows the standard evaluation protocol used in prior works for 3D molecular generation** such as EDM [1], GeoLDM [2], and RADM [3], where molecules are represented as 3D point clouds. Under this representation, discrete topological properties, such as ring structures, skeleton diversity, heteroatom patterns, chirality, QED/SA, or synthetic feasibility, cannot be defined in a model-agnostic or representation-consistent way. Computing these metrics requires reconstructing a molecular graph from 3D coordinates using heuristic bonding rules, which are not unique and can change drastically with small coordinate perturbations. As a result, **such metrics evaluate the behavior of the bond-perception heuristic rather than the generative model itself**.
>   - For this reason, existing 3D diffusion baselines do not use these topology- or pattern-based metrics, and it would not be scientifically meaningful or comparable to introduce them only for DIST. **Consistent with prior work, we report the standard geometry-based metrics (validity, stability, uniqueness), which directly assess the generated 3D structures without introducing additional ambiguity.**
>
>
> - W 5: The settings and scales of "radius r", the trade-off between quality and diversity in "replication count/perturbation intensity", the specific definition and threshold selection of pilot score s_{j} (which seems to be based on stability/legitimacy in the final draft), and the sensitivity to random seed/batch size are currently less detailed in the main text.
>   - We emphasize that **DIST is robust and not sensitive to hyperparameter choices, as demonstrated by the new ablation studies added in Appendix H** of the revised paper. These experiments vary the filtering threshold, intermediate timestep, and perturbation intensity, and the results show that DIST consistently achieves strong performance across a wide range of settings without requiring fine-tuning. For example, when applied to EDM, DIST achieves more than a **at least 5\% improvement in molecule stability** across intermediate timesteps from 200 to 500 and thresholds from 0.82 to 0.88.
>   - In the original submission, following standard practice in prior works such as EDM [1], GeoLDM [2], and RADM [3], we reported the **average and standard deviation over three runs** for the QM9 results in **Table 2**. For Geom-Drugs, the deviation was negligible after rounding and therefore omitted. DIST is not sensitive to it. To ensure a **fair comparison with baselines**, we followed the same **standard setting** as these works, generating **10,000 molecules** with a **fixed batch size of 100**.
>   - In DIST, the perturbation magnitude is constrained to remain sufficiently small so that each replicated batch lies within the radius $r$ constraint. This radius is **data-dependent**, reflecting the local geometry of the underlying molecular distribution, rather than a manually tuned hyperparameter. **Nonetheless, we include an ablation study on perturbation intensity in Appendix H.3 in the revised paper**, showing that DIST remains robust across a wide range of perturbation scales. For example, when combined with EDM, DIST achieves **at least a 7\% improvement in molecule stability** even when the perturbation varies from no noise to a Gaussian scale of 0.1.
> ### Questions
> - Q 1: Provide the total end-to-end time (including pilot cost and the overhead of dropped batches) and the equivalent GPU hours, and compare it with the baseline on the same hardware and concurrency level.
>   - The computational cost of pilot evaluation and discarded batches has already been included in all efficiency analyses in the original submission. In the revised manuscript, **Appendix G.2 additionally reports the full end-to-end wall-clock comparison** under identical hardware and concurrency settings, confirming that DIST achieves the stated reduction in overall sampling cost relative to the baselines. **This concern is closely related to W 3, and we refer the reviewer to our detailed explanation there.**

---

> > ### Author Response · Authors · 2025-11-20
> > **Response to Reviewer DXTg - Part III**
> >
> > - Q 2: The image comparison in Fig. 1 is intuitive, but could the differences be exaggerated due to variations in image task semantics, etc.? Could you provide a 1D/2D toy density comparison to more purely represent the mechanism of "multiple narrow peaks + overlap + overshoot"?
> >   - **The differences in Fig. 1 are not exaggerated.** For image data, “semantic difference’’ simply reflects how far generated samples drift from the **true data manifold**. For molecular data, the numerical scale of coordinates is not the meaningful quantity; what matters are the chemical constraints encoded in those coordinates. Thus, the visual gap between valid and invalid samples naturally reflects the underlying structural deviations rather than an artifact of the task.
> >   - We also provide **both 2D and 1D toy examples to directly illustrate the phenomena caused by DC-structure and the overshoot mechanism**. These Mixture-of-Gaussian examples and the code to generate them are included in the anonymous repository (https://anonymous.4open.science/r/Mixture-of-Gaussian-Toy-Example-3A24/README.md), offering a clean, semantics-free visualization of how DC-structure leads to overshoot and drift in diffusion sampling. Code is in the Jupyter Notebook and is easy to use and run.
> >
> > - Q 3: Examine the sensitivity differences of DIST under canonicalization/coordinate normalization schemes.
> >   - This is an interesting question - actually we've investigated canonicalization for 3D point clouds before. **Unfortunately, as far as we know, there does not exist a continuous **canonicalizer** for a generic point cloud in 3D Euclidean space,** since any coordinate canonicalization would require breaking group symmetry. The mathematical argument for the **non-existence of a canonicalizer** under such symmetry can be found in [4].
> >   - Among our baselines, RADM[3] employs a **latent-space rotational alignment mechanism**, which is similar to canonicalization. The consistent performance of DIST demonstrates the robustness and generality of our approach with respect to coordinate representations.
> > - Q 4: Please change "TV–contraction" to "TV non-expansion" and clarify that κ=1 in general. Furthermore, providing sufficient conditions under which κ<1 would be more convincing.
> >   - **This concern is closely related to W 1, and we refer the reviewer to our detailed explanation there.**  We have included in Appendix E.1 an argument establishing this stricter bound for Gaussian transitions in the revised paper rather than the general Markov kernels. Throughout the paper, the term *TV-contraction* is used in this setting precisely.
> > - Q 5: Please provide accurate accounting (including pilot costs for failed batches) in Appendix D, and supplement with wall-clock and throughput to avoid substituting actual time with just "steps".
> >   - **The accounting in the original paper had already included pilot costs for failed batches.** And we additionally provide the **wall-clock comparison in Appendix G.2** in the revised paper. **This concern is closely related to W 3, and we refer the reviewer to our detailed explanation there.**
> >
> > [1] Equivariant Diffusion for Molecule Generation in 3D, ICML 2022
> >
> > [2] Geometric Latent Diffusion Models for 3D Molecule Generation, ICML 2023
> >
> > [3] Scalable Non-Equivariant 3D Molecule Generation via Rotational Alignment, ICML 2025
> >
> > [4] Equivariant Frames and the Impossibility of Continuous Canonicalization, ICML 2024

---

> > > ### Author Response · Authors · 2025-11-28
> > > **Follow-up on Our Responses - Reviewer DXTg**
> > >
> > > Dear Reviewer DXTg,
> > >
> > > We sincerely thank you again for your thoughtful suggestions and detailed feedback. Since it has now been more than a week since we posted our response, we would like to follow up and ensure that all of your concerns have been fully addressed.
> > >
> > > - If there are any remaining questions or points that would benefit from further clarification, we would be very happy to provide additional explanation.
> > > - As reflected in our response and revised manuscript, we have added substantial new material, including:
> > >   - **Ablation studies** demonstrating the robustness of DIST under different hyperparameter settings,
> > >   - **2D and 1D toy examples** illustrating the DC-structure issue and the overshoot mechanism,
> > >   - **Experiments** validating the existence and impact of DC-structure,
> > >   - **Mathematical derivations** clarifying the TV-contraction argument,
> > >   - **Expanded discussion of related work** to clearly articulate the originality and contributions of DIST,
> > >   - **Pseudocode and an end-to-end wall-clock evaluation** to fully support our claims on efficiency.
> > >
> > > - We believe that several earlier concerns may have arisen from misunderstandings, and we have now addressed these carefully in the previous response and revised version.
> > >
> > > If there are no further issues, we would greatly appreciate it if you could kindly reconsider your rating.
> > >
> > > Thank you again for your time and constructive feedback.
> > >
> > > Best regards,
> > > **The Authors**

---

### Meta-Review · Area_Chair_SfFA · 2025-12-20

**Summary:**

The paper addresses the fragility of diffusion models in 3D molecular generation, attributed to what the authors define as Dense-Concentrated (DC) structure. Unlike images, molecules have strict chemical/geometric constraints that create sharp probability peaks. The authors propose DIST, a plug-in corrective module that evaluates trajectories at intermediate timesteps, filters out those drifting into low-density "invalid" regions, and steers the process back toward valid peaks.

The primary concerns from reviewers included the lack of a stricter theoretical bound for TV-contraction, the absence of wall-clock efficiency analysis (beyond step counts), questions regarding the novelty compared to "exposure bias" literature, and skepticism about the deterministic approximation used to derive the "overshoot" mechanism.

My own concern:

Authors claimed that DIST is explicitly designed as a model-agnostic and plug-in corrective module. This design makes it theoretically adaptable to other 3D data representations like point clouds and meshes, provided they are modeled through a diffusion process that suffers from similar distribution-level discrepancies. However, DIST may encounter scalability issue when handling massive 3D points due to its Pilot-Sample mechanism and difficulty in defining a "validity" metric for a raw point cloud. I would suggest the authors to reframe their claim or providing more evidence showing DIST's model-agnostic and plug-in features.

Besides, though the authors claimed that the proposed scheme is for "concentrated" distribution, there is no statistics or values quantifying how "concentrated" a molecule dataset is. If two datasets are concentrated at different levels, what is the corresponding impact?

**Reviewer Concerns:**

Addressed concerns:

-  Theoretical Bound: The authors successfully provided a formal proof for strict TV-contraction ($\kappa < 1$) specifically for Gaussian transitions, moving beyond the general non-expansive Markov kernel argument.
- Efficiency Analysis: The authors supplemented their step-count claims with comprehensive wall-clock time results (Appendix G.2), showing that DIST reduces end-to-end sampling time (e.g., from 145 to 83 minutes for EDM).
-  Comparison with Exposure Bias (wDAK): The authors clarified that while exposure bias focuses on local transition mismatch, DIST addresses global distributional correctness through selective sampling, achieving NFE-level efficiency without retraining.
- Ablation Studies: Extensive studies on thresholds, timesteps, and perturbation intensity were added, proving the method is robust and not overly sensitive to hyperparameter tuning.

Outstanding concerns:

- Mathematical Justification of Overshoot: Reviewer wDAK remained skeptical about the derivation near Equation 10, specifically regarding a coefficient in the score function and the validity of ignoring the stochastic term. While the authors provided Appendix C.3 to justify this as an isolation of "deterministic displacement," the reviewer remained unconvinced of its mathematical rigor.
- Scalability: Concerns about protein-level systems remain technically unproven, though the authors reasonably argued this is a fundamentally different task with different data distributions.

**Reviewer Scores:**

- DXTg (4): Likely would have moved to a 6. The reviewer’s primary technical "Weaknesses" (stricter bound and wall-clock time) were addressed with specific new proofs and data in the revision.
- wDAK (2): Likely would have moved to a 4. While the reviewer acknowledged the toy examples and domain gap findings were "quite interesting," they remained deeply skeptical of the mathematical derivation of the core overshoot mechanism.
- 50Dq (4): Likely would have moved to a 6. This reviewer's concerns were largely centered on ablation/sensitivity and related work, which were thoroughly addressed in the rebuttal and new Appendix H.

---

### Decision · Program_Chairs · 2026-01-26

Reject